# Boosting with Abstention

**Corinna Cortes**
Google Research
New York, NY 10011
corinna@google.com

**Giulia DeSalvo**
Courant Institute
New York, NY 10012
desalvo@cims.nyu.edu

**Mehryar Mohri**
Courant Institute and Google
New York, NY 10012
mohri@cims.nyu.edu

## Abstract

We present a new boosting algorithm for the key scenario of binary classification with abstention where the algorithm can abstain from predicting the label of a point, at the price of a fixed cost. At each round, our algorithm selects a pair of functions, a base predictor and a base abstention function. We define convex upper bounds for the natural loss function associated to this problem, which we prove to be calibrated with respect to the Bayes solution. Our algorithm benefits from general margin-based learning guarantees which we derive for ensembles of pairs of base predictor and abstention functions, in terms of the Rademacher complexities of the corresponding function classes. We give convergence guarantees for our algorithm along with a linear-time weak-learning algorithm for abstention stumps. We also report the results of several experiments suggesting that our algorithm provides a significant improvement in practice over two confidence-based algorithms.

## 1 Introduction

Classification with abstention is a key learning scenario where the algorithm can abstain from making a prediction, at the price of incurring a fixed cost. This is the natural scenario in a variety of common and important applications. An example is spoken-dialog applications where the system can redirect a call to an operator to avoid the cost of incorrectly assigning a category to a spoken utterance and misguiding the dialog manager. This requires the availability of an operator, which incurs a fixed and predefined price. Other examples arise in the design of a search engine or an information extraction system, where, rather than taking the risk of displaying an irrelevant document, the system can resort to the help of a more sophisticated, but more time-consuming classifier. More generally, this learning scenario arises in a wide range of applications including health, bioinformatics, astronomical event detection, active learning, and many others, where abstention is an acceptable option with some cost. Classification with abstention is thus a highly relevant problem.

The standard approach for tackling this problem is via confidence-based abstention: a real-valued function $h$ is learned for the classification problem and the points $x$ for which its magnitude $|h(x)|$ is smaller than some threshold $\gamma$ are rejected. Bartlett and Wegkamp [1] gave a theoretical analysis of this approach based on consistency. They introduced a discontinuous loss function taking into account the cost for rejection, upper-bounded that loss by a convex and continuous Double Hinge Loss (DHL) surrogate, and derived an algorithm based on that convex surrogate loss. Their work inspired a series of follow-up papers that developed both the theory and practice behind confidence-based abstention [32, 15, 31]. Further related works can be found in Appendix A.

In this paper, we present a solution to the problem of classification with abstention that radically departs from the confidence-based approach. We introduce a general model where a pair $(h, r)$ for a classifier $h$ and rejection function $r$ are learned simultaneously. Under this novel framework, we present a Boosting-style algorithm with Abstention, BA, that learns accurately the classifier and abstention functions. Note that the terminology of "boosting with abstention" was used by Schapire and Singer [26] to refer to a scenario where a base classifier is allowed to abstain, but

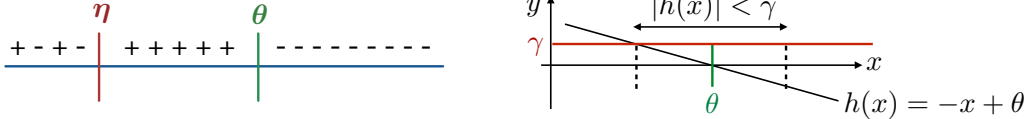

Figure 1: The best predictor $h$ is defined by the threshold $\theta$: $h(x) = -x + \theta$. For $c < \frac{1}{2}$, the region defined by $X \leq \eta$ should be rejected. But the corresponding abstention function $r$ defined by $r(x) = x - \eta$ cannot be defined as $|h(x)| \leq \gamma$ for any $\gamma > 0$.

where the boosting algorithm itself has to commit to a prediction. This is therefore distinct from the scenario of classification with abstention studied here. Nevertheless, we will introduce and examine a confidence-based Two-Step Boosting algorithm, the TSB algorithm, that consists of first training Adaboost and next searching for the best confidence-based abstention threshold.

The paper is organized as follows. Section 2 describes our general abstention model which consists of learning a pair $(h, r)$ simultaneously and compares it with confidence-based models. Section 3.2 presents a series of theoretical results for the problem of learning convex ensembles for classification with abstention, including the introduction of calibrated convex surrogate losses and general data-dependent learning guarantees. In Section 4, we use these learning bounds to design a regularized boosting algorithm. We further prove the convergence of the algorithm and present a linear-time weak-learning algorithm for a natural family of *abstention stumps*. Finally, in Section 5, we report several experimental results comparing the BA algorithm with the DHL and the TSB algorithms.

## 2 Preliminaries

In this section, we first introduce a general model for learning with abstention [7] and then compare it with confidence-based models.

### 2.1 General abstention model

We assume as in standard supervised learning that the training and test points are drawn i.i.d. according to some fixed but unknown distribution $\mathcal{D}$ over $\mathcal{X} \times \{-1, +1\}$. We consider the learning scenario of binary classification with abstention. Given an instance $x \in \mathcal{X}$, the learner has the option of abstaining from making a prediction for $x$ at the price of incurring a non-negative loss $c(x)$, or otherwise making a prediction $h(x)$ using a predictor $h$ and incurring the standard zero-one loss $1_{yh(x) \leq 0}$ where the true label is $y$. Since a random guess achieves an expected cost of at most $\frac{1}{2}$, rejection only makes sense for $c(x) < \frac{1}{2}$.

We will model the learner by a pair $(h, r)$ where the function $r \colon \mathcal{X} \to \mathbb{R}$ determines the points $x \in \mathcal{X}$ to be rejected according to $r(x) \leq 0$ and where the hypothesis $h \colon \mathcal{X} \to \mathbb{R}$ predicts labels for non-rejected points via its sign. Extending the loss function considered in Bartlett and Wegkamp [1], the abstention loss for a pair $(h, r)$ is defined as as follows for any $(x, y) \in \mathcal{X} \times \{-1, +1\}$:

$$L(h, r, x, y) = 1_{yh(x) \leq 0} 1_{r(x) > 0} + c(x) 1_{r(x) \leq 0}. \tag{1}$$

The abstention cost $c(x)$ is assumed known to the learner. In the following, we assume that $c$ is a constant function, but part of our analysis is applicable to the more general case.

We denote by $\mathcal{H}$ and $\mathcal{R}$ two families of functions mapping $\mathcal{X}$ to $\mathbb{R}$ and we assume the labeled sample $S = ((x_1, y_1), \ldots, (x_m, y_m))$ is drawn i.i.d. from $\mathcal{D}^m$. The learning problem consists of determining a pair $(h, r) \in \mathcal{H} \times \mathcal{R}$ that admits a small expected abstention loss $R(h, r)$, defined as follows:

$$R(h, r) = \mathop{\mathbb{E}}_{(x,y) \sim \mathcal{D}} \left[ 1_{yh(x) \leq 0} 1_{r(x) > 0} + c 1_{r(x) \leq 0} \right]. \tag{2}$$

Similarly, we define the empirical loss of a pair $(h, r) \in \mathcal{H} \times \mathcal{R}$ over the sample $S$ by: $\widehat{R}_S(h, r) = \mathbb{E}_{(x,y) \sim S} \left[ 1_{yh(x) \leq 0} 1_{r(x) > 0} + c 1_{r(x) \leq 0} \right]$, where $(x, y) \sim S$ indicates that $(x, y)$ is drawn according to the empirical distribution defined by $S$.

### 2.2 Confidence-based abstention model

Confidence-based models are a special case of the general model for learning with rejection presented in Section 2.1 corresponding to the pair $(h(x), r(x)) = (h(x), |h(x)| - \gamma)$, where $\gamma$ is a parameter

that changes the threshold of rejection. This specific choice was based on consistency results shown in [1]. In particular, the Bayes solution $(h^*, r^*)$ of the learning problem, that is where the distribution $\mathcal{D}$ is known, is given by $h^*(x) = \eta(x) - \frac{1}{2}$ and $r^*(x) = |h^*(x)| - (\frac{1}{2} - c)$ where $\eta(x) = \mathbb{P}[Y = +1|x]$ for any $x \in \mathcal{X}$, but note that this is not a unique solution. The form of $h^*(x)$ follows by a similar reasoning as for the standard binary classification problem. It is straightforward to see that the optimal rejection function $r^*$ is non-positive, meaning a point is rejected, if and only if $\min\{\eta(x), 1 - \eta(x)\} \geq c$. Equivalently, the following holds: $\max\{\eta(x) - \frac{1}{2}, \frac{1}{2} - \eta(x)\} \leq \frac{1}{2} - c$ if and only if $|\eta(x) - \frac{1}{2}| \leq \frac{1}{2} - c$ and using the definition of $h^*$, we recover the optimal $r^*$. In light of the Bayes solution, the specific choice of the abstention function $r$ is natural; however, requiring the abstention function $r$ to be defined as $r(x) = |h(x)| - \gamma$, for some $h \in \mathcal{H}$, is in general too restrictive when predictors are selected out of a limited subset $\mathcal{H}$ of all measurable functions over $\mathcal{X}$. Consider the example shown in Figure 1 where $\mathcal{H}$ is a family of linear functions. For this simple case, the optimal abstention region cannot be attained as a function of the best predictor $h$ while it can be achieved by allowing to learn a pair $(h, r)$. Thus, the general model for learning with abstention analyzed in Section 2.1 is both more flexible and more general.

# 3   Theoretical analysis

This section presents a theoretical analysis of the problem of learning convex ensembles for classification with abstention. We first introduce general convex surrogate functions for the abstention loss and prove a necessary and sufficient condition based on their parameters for them to be calibrated. Next we define the ensemble family we consider and prove general data-dependent learning guarantees for it based on the Rademacher complexities of the base predictor and base rejector sets.

## 3.1   Convex surrogates

We introduce two types of convex surrogate functions for the abstention loss. Observe that the abstention loss $L(h, r, x, y)$ can be equivalently expressed as $L(h, r, x, y) = \max\left(1_{yh(x) \leq 0} 1_{-r(x) < 0}, c\, 1_{r(x) \leq 0}\right)$. In view of that, since for any $f, g \in \mathbb{R}$, $\max(f, g) = \frac{f+g+|g-f|}{2} \geq \frac{f+g}{2}$, the following inequalities hold for $a > 0$ and $b > 0$:

$$
\begin{aligned}
L(h, r, x, y) &= \max\left(1_{yh(x) \leq 0} 1_{-r(x) < 0},\, c\, 1_{r(x) \leq 0}\right) \\
&\leq \max\left(1_{\max(yh(x), -r(x)) \leq 0},\, c\, 1_{r(x) \leq 0}\right) \\
&\leq \max\left(1_{\frac{yh(x) - r(x)}{2} \leq 0},\, c\, 1_{r(x) \leq 0}\right) \\
&= \max\left(1_{a\,[yh(x) - r(x)] \leq 0},\, c 1_{b\, r(x) \leq 0}\right) \\
&\leq \max\left(\Phi_1\big(a\,[r(x) - yh(x)]\big),\, c\, \Phi_2\big(-b\, r(x)\big)\right),
\end{aligned}
$$

where $u \to \Phi_1(-u)$ and $u \to \Phi_2(-u)$ are two non-increasing convex functions upper-bounding $u \to 1_{u \leq 0}$ over $\mathbb{R}$. Let $L_{\mathrm{MB}}$ be the convex surrogate defined by the last inequality above:

$$
L_{\mathrm{MB}}(h, r, x, y) = \max\left(\Phi_1\big(a\,[r(x) - yh(x)]\big),\, c\, \Phi_2\big(-b\, r(x)\big)\right), \tag{3}
$$

Since $L_{\mathrm{MB}}$ is not differentiable everywhere, we upper-bound the convex surrogate $L_{\mathrm{MB}}$ as follows: $\max\left(1_{a\,[yh(x) - r(x)] \leq 0},\, c\, 1_{b\, r(x) \leq 0}\right) \leq \Phi_1\big(a\,[r(x) - yh(x)]\big) + c\, \Phi_2\big(-b\, r(x)\big)$. Similarly, we let $L_{\mathrm{SB}}$ denote this convex surrogate:

$$
L_{\mathrm{SB}}(h, r, x, y) = \Phi_1\big(a\,[r(x) - yh(x)]\big) + c\, \Phi_2\big(-b\, r(x)\big). \tag{4}
$$

Figure 2 shows the plots of the convex surrogates $L_{\mathrm{MB}}$ and $L_{\mathrm{SB}}$ as well as that of the abstention loss.

Let $(h_L^*, r_L^*)$ denote the pair that attains the minimum of the expected loss $\mathbb{E}_{x,y}(L_{\mathrm{SB}}(h, r, x, y))$ over all measurable functions for $\Phi_1(u) = \Phi_2(u) = \exp(u)$. In Appendix F, we show that with $\eta(x) = \mathbb{P}(Y = +1|X = x)$, the pair $(h_L^*, r_L^*)$ where $h_L^* = \frac{1}{2a} \log\left(\frac{\eta}{1-\eta}\right)$ and $r_L^* = \frac{1}{a+b} \log\left(\frac{cb}{2a} \sqrt{\frac{1}{\eta(1-\eta)}}\right)$ makes $L_{\mathrm{SB}}$ a calibrated loss, meaning that the sign of the $(h_L^*, r_L^*)$ that minimizes the expected surrogate loss matches the sign of the Bayes classifier $(h^*, r^*)$. More precisely, the following holds.

**Theorem 1** (Calibration of convex surrogate). *For $a > 0$ and $b > 0$, the $\inf_{(h,r)} \mathbb{E}_{(x,y)}[L(h, r, x, y)]$ is attained at $(h_L^*, r_L^*)$ such that* $\mathrm{sign}(h^*) = \mathrm{sign}(h_L^*)$ *and* $\mathrm{sign}(r^*) = \mathrm{sign}(r_L^*)$ *if and only if* $b/a = 2\sqrt{(1-c)/c}$.

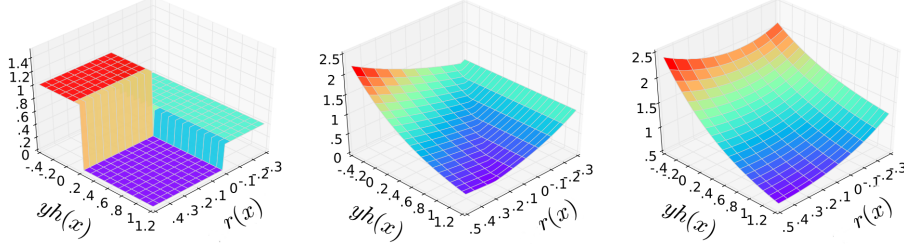

Figure 2: The left figure is a plot of the abstention loss. The middle figure is a plot of the surrogate function $L_{\mathrm{MB}}$ while the right figure is a plot of the surrogate loss $L_{\mathrm{SB}}$ both for $c = 0.45$.

The theorem shows that the classification and rejection solution obtained by minimizing the surrogate loss for that choice of $(a, b)$ coincides with the one obtained using the original loss. In the following, we make the explicit choice of $a = 1$ and $b = 2\sqrt{(1-c)/c}$ for the loss $L_{\mathrm{SB}}$ to be calibrated.

### 3.2  Learning guarantees for ensembles in classification with abstention

In the standard scenario of classification, it is often easy to come up with simple base classifiers that may abstain. As an example, a simple rule could classify a message as spam based on the presence of some word, as ham in the presence of some other word, and just abstain in the absence of both, as in the boosting with abstention algorithm by Schapire and Singer [26]. Our objective is to learn ensembles of such base hypotheses to create accurate solutions for classification with abstention. Our ensemble functions are based on the framework described in Section 2.1. Let $\mathcal{H}$ and $\mathcal{R}$ be two families of functions mapping $\mathcal{X}$ to $[-1, 1]$. The ensemble family $\mathcal{F}$ that we consider is then the convex hull of $\mathcal{H} \times \mathcal{R}$:

$$\mathcal{F} = \left\{ \left( \sum_{t=1}^{T} \alpha_t h_t, \sum_{t=1}^{T} \alpha_t r_t \right) : T \geq 1, \alpha_t \geq 0, \sum_{t=1}^{T} \alpha_t = 1, h_t \in \mathcal{H}, r_t \in \mathcal{R} \right\}. \tag{5}$$

Thus, $(\mathbf{h}, \mathbf{r}) \in \mathcal{F}$ abstains on input $x \in \mathcal{X}$ when $\mathbf{r}(x) \leq 0$ and predicts the label $\mathrm{sign}(\mathbf{h}(x))$ otherwise.

Let $u \to \Phi_1(-u)$ and $u \to \Phi_2(-u)$ be two strictly decreasing differentiable convex function upper-bounding $u \to 1_{u \leq 0}$ over $\mathbb{R}$. For calibration constants $a$, $b > 0$, and cost $c > 0$, we assume that there exist $u$ and $v$ such that $\Phi_1(a\,u) < 1$ and $c\,\Phi_2(v) < 1$, otherwise the surrogate would not be useful. Let $\Phi_1^{-1}$ and $\Phi_2^{-1}$ be the inverse functions, which always exist since $\Phi_1$ and $\Phi_2$ are strictly monotone. We will use the following definitions: $C_{\Phi_1} = 2a\,\Phi_1'\big(\Phi_1^{-1}(1)\big)$ and $C_{\Phi_2} = 2cb\,\Phi_2'\big(\Phi_2^{-1}(1/c)\big)$. Observe that for $\Phi_1(u) = \Phi_2(u) = \exp(u)$, we simply have $C_{\Phi_1} = 2a$ and $C_{\Phi_2} = 2b$.

**Theorem 2.** *Let $\mathcal{H}$ and $\mathcal{R}$ be two families of functions mapping $\mathcal{X}$ to $\mathbb{R}$. Assume $N > 1$. Then, for any $\delta > 0$, with probability at least $1 - \delta$ over the draw of a sample $S$ of size $m$ from $\mathcal{D}$, the following holds for all $(\mathbf{h}, \mathbf{r}) \in \mathcal{F}$:*

$$R(\mathbf{h}, \mathbf{r}) \leq \underset{(x,y) \sim S}{\mathbb{E}}[L_{\mathrm{MB}}(\mathbf{h}, \mathbf{r}, x, y)] + C_{\Phi_1} \mathfrak{R}_m(\mathcal{H}) + (C_{\Phi_1} + C_{\Phi_2}) \mathfrak{R}_m(\mathcal{R}) + \sqrt{\frac{\log 1/\delta}{2m}}.$$

The proof is given in Appendix C. The theorem gives effective learning guarantees for ensemble pairs $(\mathbf{h}, \mathbf{r}) \in \mathcal{F}$ when the base predictor and abstention functions admit favorable Rademacher complexities. In earlier work [7], we present a learning bound for a different type of surrogate losses which can also be extended to hold for ensembles.

Next, we derive margin-based guarantees in the case where $\Phi_1(u) = \Phi_2(u) = \exp(u)$. For any $\rho > 0$, the margin-losses associated to $L_{\mathrm{MB}}$ and $L_{\mathrm{SB}}$ are denoted by $L_{\mathrm{MB}}^{\rho}$ and $L_{\mathrm{SB}}^{\rho}$ and defined for all $(\mathbf{h}, \mathbf{r}) \in \mathcal{F}$ and $(x, y) \in \mathcal{X} \times \{-1, +1\}$ by

$$L_{\mathrm{MB}}^{\rho}(\mathbf{h}, \mathbf{r}, x, y) = L_{\mathrm{MB}}(\mathbf{h}/\rho, \mathbf{r}/\rho, x, y) \quad \text{and} \quad L_{\mathrm{SB}}^{\rho}(\mathbf{h}, \mathbf{r}, x, y) = L_{\mathrm{SB}}(\mathbf{h}/\rho, \mathbf{r}/\rho, x, y).$$

Theorem 2 applied to this margin-based loss results in the following corollary.

**Corollary 3.** *Assume $N > 1$ and fix $\rho > 0$. Then, for any $\delta > 0$, with probability at least $1 - \delta$ over the draw of an i.i.d. sample $S$ of size $m$ from $\mathcal{D}$, the following holds for all $\mathbf{f} \in \mathcal{F}$:*

$$R(\mathbf{h}, \mathbf{r}) \leq \underset{(x,y) \sim S}{\mathbb{E}}[L_{\mathrm{MB}}^{\rho}(\mathbf{h}, \mathbf{r}, x, y)] + \frac{2a}{\rho} \mathfrak{R}_m(\mathcal{H}) + \frac{2(a+b)}{\rho} \mathfrak{R}_m(\mathcal{R}) + \sqrt{\frac{\log 1/\delta}{2m}}.$$

$\mathrm{BA}(S = ((x_1, y_1), \ldots, (x_m, y_m)))$
1   **for** $i \leftarrow 1$ **to** $m$ **do**
2       $D_1(i,1) \leftarrow \frac{1}{2m}; D_1(i,2) \leftarrow \frac{1}{2m}$
3   **for** $t \leftarrow 1$ **to** $T$ **do**
4       $Z_{1,t} \leftarrow \sum_{i=1}^m D_t(i,1); Z_{2,t} \leftarrow \sum_{i=1}^m D_t(i,2)$
5       $k \leftarrow \operatorname{argmin}_{j \in [1,N]} 2Z_{1,t}\epsilon_{t,j} + Z_{1,t}\overline{r}_{j,1} - 2\sqrt{c(1-c)}Z_{2,t}\overline{r}_{j,2}$   ▷ Direction
6       $Z \leftarrow Z_{1,t}(\epsilon_{t,k} + \frac{\overline{r}_{k,1}}{2}) - 2\sqrt{c(1-c)}Z_{2,t}\frac{\overline{r}_{k,2}}{2}$
7       **if** $(Z_{1,t} - Z)e^{\alpha_{t-1,k}} - Ze^{-\alpha_{t-1,k}} < \frac{m}{Z_t}\beta$ **then**
8           $\eta_t \leftarrow -\alpha_{t-1,k}$   ▷ Step
9       **else** $\eta_t \leftarrow \log\left[-\frac{m\beta}{2Z_tZ} + \sqrt{\left[\frac{m\beta}{2Z_tZ}\right]^2 + \frac{Z_{1,t}}{Z} - 1}\right]$   ▷ Step
10      $\boldsymbol{\alpha}_t \leftarrow \boldsymbol{\alpha}_{t-1} + \eta_t \mathbf{e}_k$
11      $\mathbf{r}_t \leftarrow \sum_{j=1}^N \alpha_j r_j$
12      $\mathbf{h}_t \leftarrow \sum_{j=1}^N \alpha_j h_j$
13      $Z_{t+1} \leftarrow \sum_{i=1}^m \Phi'\big(\mathbf{r}_t(x_i) - y_i\mathbf{h}_t(x_i)\big) + \Phi'\big(-2\sqrt{\frac{1-c}{c}}\mathbf{r}_t(x_i)\big)$
14      **for** $i \leftarrow 1$ **to** $m$ **do**
15          $D_{t+1}(i,1) \leftarrow \frac{\Phi'\big(\mathbf{r}_t(x_i) - y_i\mathbf{h}_t(x_i)\big)}{Z_{t+1}}; D_{t+1}(i,2) \leftarrow \frac{\Phi'\big(-2\sqrt{\frac{1-c}{c}}\mathbf{r}_t(x_i)\big)}{Z_{t+1}}$
16  $(\mathbf{h}, \mathbf{r}) \leftarrow \sum_{j=1}^N \alpha_{T,j}(\mathbf{h}_j, \mathbf{r}_j)$
17  **return** $(\mathbf{h}, \mathbf{r})$

Figure 3: Pseudocode of the BA algorithm for both the exponential loss with $\Phi_1(u) = \Phi_2(u) = \exp(u)$ as well as for the logistic loss with $\Phi_1(u) = \Phi_2(u) = \log_2(1 + e^u)$. The parameters include the cost of rejection $c$ and $\beta$ determining the strength of the the $\boldsymbol{\alpha}$-constraint for the L1 regularization. The definition of the weighted errors $\epsilon_{t,k}$ as well as the expected rejections, $\overline{r}_{k,1}$ and $\overline{r}_{k,2}$, are given in Equation 7. For other surrogate losses, the step size $\eta_t$ is found via a line search or other numerical methods by solving $\operatorname{argmin}_\eta F(\boldsymbol{\alpha}_{t-1} + \eta\mathbf{e}_k)$.

The bound of Corollary 3 applies similarly to $L_{\mathrm{SB}}^\rho$ since it is an upper bound on $L_{\mathrm{MB}}^\rho$. It can further be shown to hold uniformly for all $\rho \in (0, 1)$ at the price of a term in $O\left(\sqrt{\frac{\log\log 1/\rho}{m}}\right)$ using standard techniques [16, 22] (see Appendix C).

## 4   Boosting algorithm

Here, we derive a boosting-style algorithm (BA algorithm) for learning an ensemble with the option of abstention for both losses $L_{\mathrm{MB}}$ and $L_{\mathrm{SB}}$. Below, we describe the algorithm for $L_{\mathrm{SB}}$ and refer the reader to Appendix H for the version using the loss $L_{\mathrm{MB}}$.

### 4.1   Objective function

The BA algorithm solves a convex optimization problem that is based on Corollary 3 for loss $L_{\mathrm{SB}}$. Since the last three terms of the right-hand side of the bound of the corollary do not depend on $\boldsymbol{\alpha}$, this suggests to select $\boldsymbol{\alpha}$ as the solution of $\min_{\boldsymbol{\alpha} \in \Delta} \frac{1}{m}\sum_{i=1}^m L_{\mathrm{SB}}^\rho(\mathbf{h}, \mathbf{r}, x_i, y_i)$. Via a change of variable $\boldsymbol{\alpha} \leftarrow \boldsymbol{\alpha}/\rho$ that does not affect the optimization problem, we can equivalently search for $\min_{\boldsymbol{\alpha} \geq 0} \frac{1}{m}\sum_{i=1}^m L_{\mathrm{SB}}(\mathbf{h}, \mathbf{r}, x_i, y_i)$ such that $\sum_{t=1}^T \alpha_t \leq 1/\rho$. Introducing the Lagrange variable $\beta$ associated to the constraint $\sum_{t=1}^T \alpha_t \leq 1/\rho$, the problem can rewritten as: $\min_{\boldsymbol{\alpha} \geq 0} \frac{1}{m}\sum_{i=1}^m L_{\mathrm{SB}}(\mathbf{h}, \mathbf{r}, x_i, y_i) + \beta\sum_{t=1}^T \alpha_t$. Letting $\{(h_1, r_1), \ldots, (h_N, r_N)\}$ be the set of base functions pairs for the classifier and rejection function, we can rewrite the optimization problem as

the minimization over $\boldsymbol{\alpha} \geq 0$ of

$$\frac{1}{m}\sum_{i=1}^{m}\Phi\Big(\sum_{j=1}^{N}\alpha_j r_j(x_i)-y_i\sum_{j=1}^{N}\alpha_j h_j(x_i)\Big)+c\,\Phi\Big(-b\sum_{j=1}^{N}\alpha_j r_j(x_i)\Big)+\beta\sum_{j=1}^{N}\alpha_j.$$

Thus, the following is the objective function of our optimization problem:

$$F(\boldsymbol{\alpha}) = \frac{1}{m}\sum_{i=1}^{m}\Phi\big(\mathbf{r}_t(x_i)-y_i\mathbf{h}_t(x_i)\big) + c\,\Phi\big(-b\,\mathbf{r}_t(x_i)\big) + \beta\sum_{j=1}^{N}\alpha_j. \tag{6}$$

## 4.2 Projected coordinate descent

The problem $\min_{\boldsymbol{\alpha}\geq 0} F(\boldsymbol{\alpha})$ is a convex optimization problem, which we solve via projected coordinate descent. Let $\mathbf{e}_k$ be the $k$th unit vector in $\mathbb{R}^N$ and let $F'(\boldsymbol{\alpha},\mathbf{e}_j)$ be the directional derivative of $F$ along the direction $\mathbf{e}_j$ at $\boldsymbol{\alpha}$. The algorithm consists of the following three steps. First, it determines the direction of maximal descent by $k = \operatorname{argmax}_{j\in[1,N]}|F'(\boldsymbol{\alpha}_{t-1},\mathbf{e}_j)|$. Second, it calculates the best step $\eta$ along the direction that preserves non-negativity of $\boldsymbol{\alpha}$ by $\eta = \operatorname{argmin}_{\boldsymbol{\alpha}_{t-1}+\eta\mathbf{e}_k\geq 0} F(\boldsymbol{\alpha}_{t-1}+\eta\mathbf{e}_k)$. Third, it updates $\boldsymbol{\alpha}_{t-1}$ to $\boldsymbol{\alpha}_t = \boldsymbol{\alpha}_{t-1}+\eta\mathbf{e}_k$.

The pseudocode of the BA algorithm is given in Figure 3. The step and direction are based on $F'(\boldsymbol{\alpha}_{t-1},\mathbf{e}_j)$. For any $t\in[1,T]$, define a distribution $D_t$ over the pairs $(i,n)$, with $n$ in $\{1,2\}$

$$D_t(i,1) = \frac{\Phi'\big(\mathbf{r}_{t-1}(x_i)-y_i\mathbf{h}_{t-1}(x_i)\big)}{Z_t} \quad \text{and} \quad D_t(i,2) = \frac{\Phi'\big(-b\,\mathbf{r}_{t-1}(x_i)\big)}{Z_t},$$

where $Z_t$ is the normalization factor given by $Z_t = \sum_{i=1}^{m}\Phi'\big(\mathbf{r}_{t-1}(x_i)-y_i\mathbf{h}_{t-1}(x_i)\big) + \Phi'\big(-b\,\mathbf{r}_{t-1}(x_i)\big)$. In order to derive an explicit formulation of the descent direction that is based on the weighted error of the classification function $h_j$ and the expected value of the rejection function $r_j$, we use the distributions $D_{1,t}$ and $D_{2,t}$ defined by $D_t(i,1)/Z_{1,t}$ and $D_t(i,1)/Z_{2,t}$ where $Z_{1,t} = \sum_{i=1}^{m} D_t(i,1)$ and $Z_{2,t} = \sum_{i=1}^{m} D_t(i,2)$ are the normalization factors. Now, for any $j\in[1,N]$ and $s\in[1,T]$, we can define the weighted error $\epsilon_{t,j}$ and the expected value of the rejection function, $\bar{r}_{j,1}$ and $\bar{r}_{j,2}$, over distribution $D_{1,t}$ and $D_{2,t}$ as follows:

$$\epsilon_{t,j} = \frac{1}{2}\Big[1 - \operatorname*{\mathbb{E}}_{i\sim D_{1,t}}[y_i h_j(x_i)]\Big], \quad \bar{r}_{j,1} = \operatorname*{\mathbb{E}}_{i\sim D_{1,t}}[r_j(x_i)], \text{ and } \bar{r}_{j,2} = \operatorname*{\mathbb{E}}_{i\sim D_{2,t}}[r_j(x_i)]. \tag{7}$$

Using these definition, we show (see Appendix D) that the descent direction is given by

$$k = \operatorname*{argmin}_{j\in[1,N]} 2Z_{1,t}\epsilon_{t,j} + Z_{1,t}\bar{r}_{j,1} - 2\sqrt{c(1-c)}Z_{2,t}\bar{r}_{j,2}.$$

This equation shows that $Z_{1,t}$ and $2\sqrt{c(1-c)}Z_{2,t}$ re-scale the weighted error and expected rejection. Thus, finding the best descent direction by minimizing this equation is equivalent to finding the best scaled trade-off between the misclassification error and the average rejection cost. The step size can in general be found via line search or other numerical methods, but we have derived a closed-form solution of the step size for both the exponential and logistic loss (see Appendix D.2). Further details of the derivation of projected coordinate descent on $F$ are also given in Appendix D.

Note that for $\mathbf{r}_t \to 0^+$ in Equation 6, that is when the rejection terms are dropped in the objective, we retrieve the L1-regularized Adaboost. As for Adaboost, we can define a *weak learning assumption* which requires that the directional derivative along at least one base pair be non-zero. For $\beta = 0$, it does not hold when for all $j$: $2\epsilon_{s,j} - 1 = -\bar{r}_{j,1} + \frac{2\sqrt{c(1-c)}Z_{2,t}}{Z_{1,t}}\bar{r}_{j,2}$, which corresponds to a balance between the edge and rejection costs for all $j$. Observe that in the particular case when the rejection functions are zero, it coincides with the standard weak learning assumption for Adaboost ($\epsilon_{s,j} = \frac{1}{2}$ for all $j$).

The following theorem provides the convergence of the projected coordinate descent algorithm for our objective function, $F(\boldsymbol{\alpha})$. The proof is given in Appendix E.

**Theorem 4.** *Assume that $\Phi$ is twice differentiable and that $\Phi''(u) > 0$ for all $u\in\mathbb{R}$. Then, the projected coordinate descent algorithm applied to $F$ converges to the solution $\boldsymbol{\alpha}^*$ of the optimization problem $\max_{\boldsymbol{\alpha}\geq 0} F(\boldsymbol{\alpha})$. If additionally $\Phi$ is strongly convex over the path of the iterates $\boldsymbol{\alpha}_t$ then there exists $\tau > 0$ and $\nu > 0$ such that for all $t > \tau$, $F(\boldsymbol{\alpha}_{t+1}) - F(\boldsymbol{\alpha}^*) \leq \big(1-\frac{1}{\nu}\big)\big(F(\boldsymbol{\alpha}_t)-F(\boldsymbol{\alpha}^*)\big)$.*

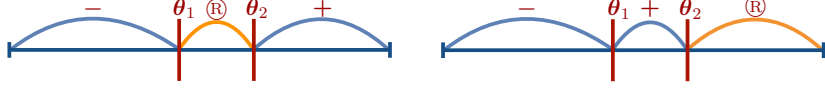

Figure 4: Illustration of the abstention stumps on a variable $X$.

Specifically, this theorem holds for the exponential loss $\Phi(u) = \exp(u)$ and the logistic loss $\Phi(-u) = \log_2(1 + e^{-u})$ since they are strongly convex over the compact set containing the $\alpha_t$s.

### 4.3 Abstention stumps

We first define a family of base hypotheses, *abstention stumps*, that can be viewed as extensions of the standard boosting stumps to the setting of classification with abstention. An abstention stump $h_{\theta_1,\theta_2}$ over the feature $X$ is defined by two thresholds $\theta_1, \theta_2 \in \mathbb{R}$ with $\theta_1 \leq \theta_2$. There are 6 different such stumps, Figure 4 illustrates two of them. For the left figure, points with variables $X$ less than or equal to $\theta_1$ are labeled negatively, those with $X \geq \theta_2$ are labeled positively, and those with $X$ between $\theta_1$ and $\theta_2$ are rejected. In general, an abstention stump is defined by the pair $\left(h_{\theta_1,\theta_2}(X), r_{\theta_1,\theta_2}(X)\right)$ where, for Figure 4-left, $h_{\theta_1,\theta_2}(X) = -1_{X \leq \theta_1} + 1_{X > \theta_2}$ and $r_{\theta_1,\theta_2}(X) = 1_{\theta_1 < X \leq \theta_2}$.

Thus, our abstention stumps are pairs $(h, \hat{r})$ with $h$ taking values in $\{-1, 0, 1\}$ and $\hat{r}$ in $\{0, 1\}$, and such that for any $x$ either $h(x)$ or $\hat{r}(x)$ is zero. For our formulation and algorithm, these stumps can be used in combination with any $\gamma > 0$, to define a family of base predictor and base rejector pairs of the form $(h(x), \gamma - \hat{r}(x))$. Since $\alpha_t$ is non-negative, the value $\gamma$ is needed to correct for over-rejection by previously selected abstention stumps. The $\gamma$ can be automatically learned by adding to the set of base pairs the constant functions $(h_0, r_0) = (0, -1)$. An ensemble solution returned by the BA algorithm is therefore of the form $\left(\sum_t \alpha_t h_t(x), \sum_t \alpha_t r_t(x)\right)$ where $\alpha_t$s are the weights assigned to each base pair.

Now, consider a sample of $m$ points sorted by the value of $X$, which we denote by $X_1 \leq \cdots \leq X_m$. For abstention stumps, the derivative of the objective, $F$, can be further simplified (see Appendix G) such that the problem can be reduced to finding an abstention stump with the minimal expected abstention loss $l(\theta_1, \theta_2)$, that is

$$\underset{\theta_1,\theta_2}{\text{argmin}} \sum_{i=1}^{m} 2D_t(i,1)[1_{y_i=+1}1_{X_i \leq \theta_1} + 1_{y_i=-1}1_{X_i > \theta_2}] + \left(2D_t(i,1) - cb\,D_t(i,2)\right)1_{\theta_1 < X_i \leq \theta_2}.$$

Notice that given $m$ points, at most $(m + 1)$ thresholds need to be considered for $\theta_1$ and $\theta_2$. Hence, a straightforward algorithm inspects all possible $O(m^2)$ pairs $(\theta_1, \theta_2)$ with $\theta_1 \leq \theta_2$ in time $O(m^2)$. However, Lemma 5 below and further derivations in Appendix G, allows for an $O(m)$-time algorithm for finding optimal abstention stumps when the problem is solved without the constraint $\theta_1 \leq \theta_2$. Note that while we state the lemma for the abstention stump in Figure 4-left, similar results hold for any of the 6 types of stumps.

**Lemma 5.** *The optimization problem without the constraint* $(\theta_1 < \theta_2)$ *can be decomposed as follows:*

$$\underset{\theta_1,\theta_2}{\text{argmin}}\, l(\theta_1, \theta_2) = \underset{\theta_1}{\text{argmin}} \sum_{i=1}^{m} 2D_t(i,1)1_{y_i=+1}1_{X_i \leq \theta_1} + \left(2D_t(i,1) - cb\,D_t(i,2)\right)1_{\theta_1 < X_i} \quad (8)$$

$$+ \underset{\theta_2}{\text{argmin}} \sum_{i=1}^{m} 2D_t(i,1)1_{y_i=-1}1_{X_i > \theta_2} + \left(2D_t(i,1) - cb\,D_t(i,2)\right)1_{X_i \leq \theta_2}. \quad (9)$$

The optimization Problems (8) and (9) can be solved in linear time, via a method similar to that of finding the optimal threshold for a standard zero-one loss boosting stump. When the condition $\theta_1 < \theta_2$ does not hold, we can simply revert to finding the minimum of $l(\theta_1, \theta_2)$ in the naive way. In practice, we find most often that the optimal solution of Problem 8 and Problem 9 satisfies $\theta_1 < \theta_2$.

## 5 Experiments

In this section, we present the results of experiments with our abstention stump BA algorithm based on $L_{\text{SB}}$ for several datasets. We compare the BA algorithm with the DHL algorithm [1], as well as a

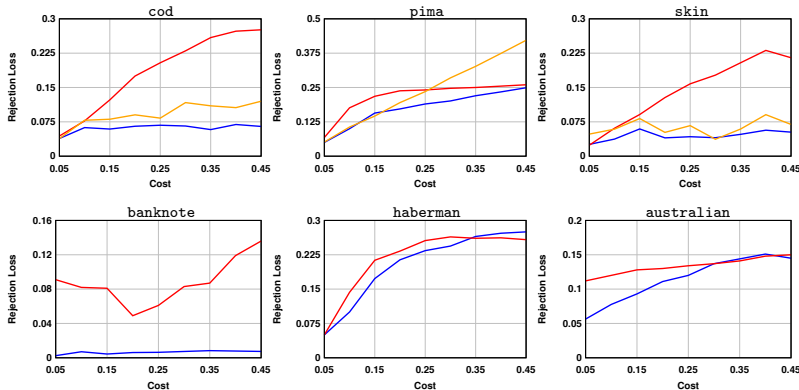

Figure 5: Average rejection loss on the test set as a function of the abstention cost $c$ for the TSB Algorithm (in orange), the DHL Algorithm (in red) and the BA Algorithm (in blue) based on $L_{\mathrm{SB}}$.

confidence-based boosting algorithm TSB. Both of these algorithms are described in further detail in Appendix B. We tested the algorithms on six data sets from UCI's data repository, specifically `australian`, `cod`, `skin`, `banknote`, `haberman`, and `pima`. For more information about the data sets, see Appendix I. For each data set, we implemented the standard 5-fold cross-validation where we randomly divided the data into training, validation and test set with the ratio 3:1:1. Using a different random partition, we repeated the experiments five times. For all three algorithms, the cost values ranged over $c \in \{0.05, 0.1, \ldots, 0.5\}$ while threshold $\gamma$ ranged over $\gamma \in \{0.08, 0.16, \ldots, 0.96\}$. For the BA algorithm, the $\beta$ regularization parameter ranged over $\beta \in \{0, 0.05, \ldots, 0.95\}$. All experiments for BA were based on $T = 200$ boosting rounds. The DHL algorithm used polynomial kernels with degree $d \in \{1, 2, 3\}$ and it was implemented in CVX [8]. For each cost $c$, the hyperparameter configuration was chosen to be the set of parameters that attained the smallest average rejection loss on the validation set. For that set of parameters we report the results on the test set.

We first compared the confidence-based TSB algorithm with the BA and DHL algorithms (first row of Figure 5). The experiments show that, while TSB can sometimes perform better than DHL, in a number of cases its performance is dramatically worse as a function of $c$ and, in all cases it is outperformed by BA. In Appendix J, we give the full set of results for the TSB algorithm.

In view of that, our next series of results focus on the BA and DHL algorithms, directly designed to optimize the rejection loss, for 3 other datasets (second row of Figure 5). Overall, the figures show that BA outperforms the state-of-the-art DHL algorithm for most values of $c$, thereby indicating that BA yields a significant improvement in practice. We have also successfully run BA on the CIFAR-10 data set (boat and horse images) which contains 10,000 instances and we believe that our algorithm can scale to much larger datasets. In contrast, training DHL on such larger samples did not terminate as it is based on a costly QCQP. In Appendix J, we present tables that report the average and standard deviation of the abstention loss as well as the fraction of rejected points and the classification error on non-rejected points.

## 6   Conclusion

We introduced a general framework for classification with abstention where the predictor and abstention functions are learned simultaneously. We gave a detailed study of ensemble learning within this framework including: new surrogate loss functions proven to be calibrated, Rademacher complexity margin bounds for ensemble learning of the pair of predictor and abstention functions, a new boosting-style algorithm, the analysis of a natural family of base predictor and abstention functions, and the results of several experiments showing that BA algorithm yield a significant improvement over the confidence-based algorithms DHL and TSB. Our algorithm can be further extended by considering more complex base pairs such as more general ternary decision trees with rejection leaves. Moreover, our theory and algorithm can be generalized to the scenario of multi-class classification with abstention, which we have already initiated.

### Acknowledgments

This work was partly funded by NSF CCF-1535987 and IIS-1618662.

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
