[Supplementary Material]

# A    Extended Related Work

Initial work in learning with abstention has focused on the optimal trade-off between the error and abstention rate [5, 6] as well as finding the optimal abstention rule based on the ROC curve [14, 28, 25]. Another series of papers used rejection options to reduce misclassification rate, but theoretical learning guarantees were not given [13, 24, 2, 17, 21]. More recently, El-Yaniv and Wiener [10, 11] study the trade-off between the coverage and the accuracy of classifiers by using an approach related to active learning.

A seemingly connected framework is that of cost-sensitive learning where the cost of misclassifying class $y_1$ as class $y_2$ may depend on the pair $(y_1, y_2)$ [12]. It would be tempting to view classification with abstention as a special instance of cost-sensitive learning with the set of classes $\{-1, +1, \circledR\}$, with $\circledR$ standing for abstention and where a different cost would be assigned to abstention. However, in our problem, $\circledR$ is not an intrinsic class: training or test samples bear no $\circledR$ label. Instead, the distribution over that set will depend on the algorithm. Thus, classification with abstention cannot be cast as a special case of cost-sensitive learning. Sequential learning with a budget is also a marginally related task where abstention functions are learned. But, unlike our approach, it is done in a two-step process where the classifier function is fixed [29, 30]. Lastly, the option of abstaining has been analyzed in related topics including the multi-class setting [27, 9, 3], reinforcement learning [19], online learning [33] and active learning [4].

# B    Confidence-based abstention model

In this appendix, we describe two confidence-based abstention algorithms: the DHL algorithm and the TSB algorithm.

## B.1    DHL algorithm

The DHL algorithm found in [1] is based on a double hinge loss, which is a hinge-type convex surrogate, with favorable consistency results. The optimization problem solved by the DHL algorithm minimizes this surrogate loss along with the constraint that the norm of the classifier is bounded by $1 - c$. More precisely, let $\mathcal{H}$ be a hypotheses sets defined in terms of PSD kernels $K$ over $\mathcal{X}$ where $\Phi$ is the feature mapping associated to $K$, then the DHL solves the following QCQP optimization problem

$$\min_{\boldsymbol{\alpha}, \boldsymbol{\xi}, \boldsymbol{\beta}} \sum_{i=1}^{m} \xi_i + \frac{1 - 2c}{c} \beta_i$$
$$\text{subject to } \sum_{i,j=1}^{m} \alpha_i \alpha_j K(x_i, x_j) \leq (1 - c)^2$$
$$\xi_i \geq 1 - y_i \left( \sum_{i=1}^{m} \alpha_i K(x_i, x) \right) \wedge \xi_i \geq 0,$$
$$\beta_i \geq -y_i \left( \sum_{i=1}^{m} \alpha_i K(x_i, x) \right) \wedge \beta_i \geq 0, i \in [1, m].$$

## B.2    Two-step Adaboost (TSB)

The TSB algorithm is a confidence-based algorithm that proceeds in two steps. The first step consists of training a vanilla Adaboost algorithm which returns a classifier $h$. Then, given classifier $h$, the second step is to search for the best threshold $\gamma$ that minimizes the empirical abstention loss. More precisely, we pick the parameter $\gamma$ via cross-validation, by choosing the threshold that minimizes the empirical abstention loss on the validation set. This is a natural confidence-based boosting algorithm and since the BA algorithm is based on boosting, it provides a useful baseline for our experiments. We implemented this algorithm using scikit-learn [23].

# C    Theoretical guarantees

In this appendix, we provide the proof of the theoretical guarantees presented in Section 3.2.

Let $u \to \Phi_1(-u)$ and $u \to \Phi_2(-u)$ be two strictly non-increasing differentiable convex functions upper-bounding $u \to 1_{u \leq 0}$ over $\mathbb{R}$. We assume that $a, b > 0$, and $c > 0$. We will use the quantity $\min(\Phi_1(a\,u), 1)$ and so we assume that there exists $u$ such that $\Phi_1(a\,u) < 1$ and similarly, we need to analyze $\min(c\Phi_2(u), 1)$ and so we assume there exists a $u$ such that $c\Phi_2(u) < 1$. Note that if these two assumptions did not hold, then the surrogate would not be useful. Let $\Phi_1^{-1}$ and $\Phi_2^{-1}$ be the inverse functions, which always exist since $\Phi_1$ and $\Phi_2$ are strictly monotone functions. For simplicity, we define $C_{\Phi_1} = 2a\,\Phi_1'(\Phi_1^{-1}(1))$ and $C_{\Phi_2} = 2cb\,\Phi_2'(\Phi_2^{-1}(1/c))$.

**Theorem 2.** *Let $\mathcal{H}$ and $\mathcal{R}$ be family of functions mapping $\mathcal{X}$ to $\mathbb{R}$. Assume $N > 1$. Then, for any $\delta > 0$, with probability at least $1 - \delta$ over the draw of a sample $S$ of size $m$ from $\mathcal{D}$, the following holds for all $(\mathbf{h}, \mathbf{r}) \in \mathcal{F}$:*

$$R(\mathbf{h}, \mathbf{r}) \leq \mathop{\mathbb{E}}_{(x,y) \sim S}[L_{\mathrm{MB}}(\mathbf{h}, \mathbf{r}, x, y)] + C_{\Phi_1}\mathfrak{R}_m(\mathcal{H}) + (C_{\Phi_1} + C_{\Phi_2})\mathfrak{R}_m(\mathcal{R}) + \sqrt{\frac{\log\frac{1}{\delta}}{2m}}.$$

*Proof.* Let $\mathcal{L}_{\mathrm{MB}, \mathcal{F}}$ be the family of functions defined by $\mathcal{L}_{\mathrm{MB}, \mathcal{F}} = \{(x, y) \mapsto \min(L_{\mathrm{MB}}(\mathbf{h}, \mathbf{r}, x, y), 1), (\mathbf{h}, \mathbf{r}) \in \mathcal{F}\}$. Since $\min(L_{\mathrm{MB}}, 1)$ is bounded by one, by the general Rademacher complexity generalization bound [16], with probability at least $1 - \delta$ over the draw of a sample $S$, the following holds:

$$R(\mathbf{h}, \mathbf{r}) \leq \mathop{\mathbb{E}}_{(x,y) \sim \mathcal{D}}[\min(L_{\mathrm{MB}}(\mathbf{h}, \mathbf{r}, x, y), 1)]$$

$$\leq \mathop{\mathbb{E}}_{(x,y) \sim S}[\min(L_{\mathrm{MB}}(\mathbf{h}, \mathbf{r}, x, y), 1)] + 2\mathfrak{R}_m(\mathcal{L}_{\mathrm{MB}, \mathcal{F}}) + \sqrt{\frac{\log\frac{1}{\delta}}{2m}}$$

$$\leq \mathop{\mathbb{E}}_{(x,y) \sim S}[L_{\mathrm{MB}}(\mathbf{h}, \mathbf{r}, x, y)] + 2\mathfrak{R}_m(\mathcal{L}_{\mathrm{MB}, \mathcal{F}}) + \sqrt{\frac{\log\frac{1}{\delta}}{2m}}.$$

Since for any $a, b \in \mathbb{R}$, $\min(\max(a, b), 1) = \max(\min(a, 1), \min(b, 1))$, we can write

$$\min(L_{\mathrm{MB}}(\mathbf{h}, \mathbf{r}, x, y), 1)$$
$$= \max\Big(\min\Big(\Phi_1\big(a\,[\mathbf{r}(x) - y\mathbf{h}(x)]\big), 1\Big), \min\Big(c\,\Phi_2\big(-b\,\mathbf{r}(x)\big), 1\Big)\Big)$$
$$\leq \min\Big(\Phi_1\big(b\,[\mathbf{r}(x) - y\mathbf{h}(x)]\big), 1\Big) + \min\Big(c\,\Phi_2\big(-b\,\mathbf{r}(x)\big), 1\Big).$$

The function $\Phi_1(a\,u)$ has a non-negative increasing derivative because it is a strictly increasing convex function. Since $\min(\Phi_1(a\,u), 1) = \Phi_1(a\,u)$ for $a\,u \leq \Phi_1^{-1}(1)$, the Lipschitz constant of $u \mapsto \min(\Phi_1(a\,u), 1)$ is given by $a\,\Phi_1'(\Phi_1^{-1}(1))$. Similarly, $u \mapsto \min(c\Phi_2(b\,u), 1)$ is also $cb\,\Phi_2'(\Phi_2^{-1}(1/c))$-Lipschitz. Then, by Talagrand's lemma [18],

$$\mathfrak{R}_m(\mathcal{L}_{\mathrm{MB}, \mathcal{F}}) \leq a\,\Phi_1'(\Phi_1^{-1}(1))\mathfrak{R}_m\big((x, y) \mapsto \mathbf{r}(x) - y\mathbf{h}(x) \colon (\mathbf{h}, \mathbf{r}) \in \mathcal{F}\big)$$
$$+ c\,b\,\Phi_2'(\Phi_2^{-1}(1/c))\,\mathfrak{R}_m\big((x, y) \mapsto -\mathbf{r}(x) \colon (\mathbf{h}, \mathbf{r}) \in \mathcal{F}\big). \tag{10}$$

We examine each of the terms in the right-hand side of the inequality:

$$\mathfrak{R}_m\big((x, y) \mapsto \mathbf{r}(x) - y\mathbf{h}(x) \colon (\mathbf{h}, \mathbf{r}) \in \mathcal{F}\big) = \mathop{\mathbb{E}}_{\boldsymbol{\sigma}}\Bigg[\sup_{(\mathbf{h}, \mathbf{r}) \in \mathcal{F}} \frac{1}{m}\sum_{i=1}^{m}\sigma_i(\mathbf{r}(x_i) - y_i\mathbf{h}(x_i))\Bigg]$$

$$\leq \mathop{\mathbb{E}}_{\boldsymbol{\sigma}}\Bigg[\sup_{(\mathbf{h}, \mathbf{r}) \in \mathcal{F}} \frac{1}{m}\sum_{i=1}^{m}\sigma_i\mathbf{r}(x_i)\Bigg] + \mathop{\mathbb{E}}_{\boldsymbol{\sigma}}\Bigg[\sup_{(\mathbf{h}, \mathbf{r}) \in \mathcal{F}} \frac{1}{m}\sum_{i=1}^{m}-\sigma_i(y_i\mathbf{h}(x_i))\Bigg]$$

$$= \mathop{\mathbb{E}}_{\boldsymbol{\sigma}}\Bigg[\sup_{(\mathbf{h}, \mathbf{r}) \in \mathcal{F}} \frac{1}{m}\sum_{i=1}^{m}\sigma_i\mathbf{r}(x_i)\Bigg] + \mathop{\mathbb{E}}_{\boldsymbol{\sigma}}\Bigg[\sup_{(\mathbf{h}, \mathbf{r}) \in \mathcal{F}} \frac{1}{m}\sum_{i=1}^{m}\sigma_i\mathbf{h}(x_i)\Bigg]$$

$$= \mathfrak{R}_m(\mathcal{R}) + \mathfrak{R}_m(\mathcal{H}),$$

since $-y_i\sigma_i$ and $\sigma_i$ are distributed in the same way, we effectively can absorb $-y_i$ into the definition of $\sigma_i$. Lastly, since the $\boldsymbol{\alpha}$ does not affect the Rademacher complexity, we have that

$\mathbb{E}_{\boldsymbol{\sigma}}\left[\sup_{(\mathbf{h},\mathbf{r})\in\mathcal{F}}\frac{1}{m}\sum_{i=1}^{m}\sigma_i\mathbf{h}(x_i)\right] = \mathfrak{R}_m(\mathcal{H})$ and similarly $\mathbb{E}_{\boldsymbol{\sigma}}\left[\sup_{(\mathbf{h},\mathbf{r})\in\mathcal{F}}\frac{1}{m}\sum_{i=1}^{m}\sigma_i\mathbf{r}(x_i)\right] = \mathfrak{R}_m(\mathcal{R})$. By a similar reasoning, we also have that

$$\mathfrak{R}_m\big((x,y)\mapsto -\mathbf{r}(x)\colon (\mathbf{h},\mathbf{r})\in\mathcal{F}\big) = \mathbb{E}_{\boldsymbol{\sigma}}\left[\sup_{(\mathbf{h},\mathbf{r})\in\mathcal{F}}\frac{1}{m}\sum_{i=1}^{m}\sigma_i\mathbf{r}(x_i)\right] = \mathfrak{R}_m(\mathcal{R}).$$

Combining the above, we have that the right-hand side of Inequality 10 is bounded as follows

$$\mathfrak{R}_m(\mathcal{L}_{\mathrm{MB},\mathcal{F}}) \leq a\ \Phi_1'(\Phi_1^{-1}(1))\mathfrak{R}_m(\mathcal{H}) + (c\,b\ \Phi_2'\big(\Phi_2^{-1}(1/c)\big) + a\ \Phi_1'(\Phi_1^{-1}(1)))\mathfrak{R}_m(\mathcal{R}),$$

which completes the proof. $\qquad\square$

By taking $\Phi_1(u) = \Phi_2(u) = \exp(u)$, we have the following theorem since in this case, we simply have that $C_{\Phi_1} = 2a$ and $C_{\Phi_2} = 2b$.

**Theorem 6.** *Let $\mathcal{H}$ and $\mathcal{R}$ be family of functions mapping $\mathcal{X}$ to $\mathbb{R}$. Assume $N > 1$. Then, for any $\delta > 0$, with probability at least $1 - \delta$ over the draw of a sample $S$ of size $m$ from $\mathcal{D}$, the following holds for all $(\mathbf{h},\mathbf{r}) \in \mathcal{F}$:*

$$R(\mathbf{h},\mathbf{r}) \quad \leq \quad \mathbb{E}_{(x,y)\sim S}[L_{\mathrm{MB}}(\mathbf{h},\mathbf{r},x,y)] \ + \ 2a\,\mathfrak{R}_m(\mathcal{H}) \ + \ 2(a\ +\ b\,)\mathfrak{R}_m(\mathcal{R}) \ + \ \sqrt{\frac{\log 1/\delta}{2m}}.$$

The corollary below is a direct consequence of the above Theorem 6 and it presents margin-based guarantees that are subsequently used to derive the BA algorithm.

**Corollary 3.** *Assume $N > 1$ and fix $\rho > 0$. Then, for any $\delta > 0$, with probability at least $1 - \delta$ over the draw of an i.i.d. sample $S$ of size $m$ from $\mathcal{D}$, the following holds for all $(\mathbf{h},\mathbf{r}) \in \mathcal{F}$:*

$$R(\mathbf{h},\mathbf{r}) \leq \mathbb{E}_{(x,y)\sim S}[L_{\mathrm{MB}}^{\rho}(\mathbf{h},\mathbf{r},x,y)] + \frac{2a}{\rho}\mathfrak{R}_m(\mathcal{H}) + \frac{2(a+b)}{\rho}\mathfrak{R}_m(\mathcal{R}) + \sqrt{\frac{\log 1/\delta}{2m}}.$$

## D   Direction and step of projected coordinate descent

In this appendix, we provide the details of the projected coordinate descent, projected CD, algorithm by first deriving the direction and then the optimal step. We give a closed form solution of the step size for exponential loss $\Phi(u) = \exp(u)$ and logistic loss $\Phi(u) = \log_2(1 + e^u)$.

### D.1   Direction

At each iteration $t-1$, the direction $\mathbf{e}_k$ selected by projected CD is $k = \mathrm{argmax}_{j\in[1,N]}|F'(\boldsymbol{\alpha}_{t-1},\mathbf{e}_j)|$ where the derivative is given by the following

$$F'(\boldsymbol{\alpha}_{t-1},\mathbf{e}_j) = \frac{1}{m}\sum_{i=1}^{m}\Big([r_j(x_i) - y_i h_j(x_i)]\Phi'\big(\mathbf{r}_{t-1}(x_i) - y_i\mathbf{h}_{t-1}(x_i)\big)$$
$$- cb\, r_j(x_i)\Phi'\big(-b\,\mathbf{r}_{t-1}(x_i)\big)\Big) + \beta.$$

Using the definition of $D(i,1)$ and $D(i,2)$, we re-write the derivative as follows:

$$F'(\boldsymbol{\alpha}_{t-1},\mathbf{e}_j) = \frac{Z_t}{m}\sum_{i=1}^{m}\Big([r_j(x_i) - y_i h_j(x_i)]D_t(i,1) - cb\, r_j(x_i)D_t(i,2)\Big) + \beta$$
$$= \frac{Z_t}{m}\Big(2Z_{1,t}\epsilon_{s,j} - Z_{1,t} + Z_{1,t}\overline{r}_{j,1} - cb\, Z_{2,t}\overline{r}_{j,2}\Big) + \beta.$$

Hence, we have that the descent direction is $k = \mathrm{argmin}_{j\in[1,N]}\, 2Z_{1,t}\epsilon_{t,j} + Z_{1,t}\overline{r}_{j,1} - cb\, Z_{2,t}\overline{r}_{j,2}$.

## D.2 Step

The optimal step values $\eta$ for direction $\mathbf{e}_k$ is given by $\text{argmin}_{\eta+\alpha_{t-1,k}\geq 0} F(\boldsymbol{\alpha}_{t-1}+\eta\mathbf{e}_k)$. The values $\eta$ may be found via line search or other numerical methods, but below we derive a closed-form solution by minimizing an upper bound of $F(\boldsymbol{\alpha}_{t-1}+\eta\mathbf{e}_k)$.

Since $\Phi$ is convex and since for all $i \in [1,m]$

$$-y_i h_k(x_i) + r_k(x_i) = \frac{1+y_i h_k(x_i) - r_k(x_i)}{2} \cdot (-1) + \frac{1-y_i h_k(x_i) + r_k(x_i)}{2} \cdot (1),$$

we have that the following holds for all $\eta \in \mathbb{R}$

$$\Phi\big(\mathbf{r}_{t-1}(x_i) - y_i \mathbf{h}_{t-1}(x_i) - \eta y_i h_k(x_i) + \eta r_k(x_i)\big)$$
$$\leq \frac{1+y_i h_k(x_i) - r_k(x_i)}{2} \Phi\big(\mathbf{r}_{t-1}(x_i) - y_i \mathbf{h}_{t-1}(x_i) - \eta\big)$$
$$+ \frac{1-y_i h_k(x_i) + r_k(x_i)}{2} \Phi\big(\mathbf{r}_{t-1}(x_i) - y_i \mathbf{h}_{t-1}(x_i) + \eta\big).$$

Similarly, we have that $-b\, r_k(x_i) = \frac{-b\, r_k(x_i)}{2} \cdot (1) + \frac{b\, r_k(x_i)}{2} \cdot (-1)$

$$\Phi\big(-b\,\mathbf{r}_{t-1}(x_i) - b\,\eta r_k(x_i)\big)$$
$$\leq \frac{-b\, r_k(x_i)}{2} \Phi\big(-b\,\mathbf{r}_{t-1}(x_i) + \eta\big) + \frac{b\, r_k(x_i)}{2} \Phi\big(-b\,\mathbf{r}_{t-1}(x_i) - \eta\big)$$

Thus, we can upper-bound $F$ as follows:

$$F(\boldsymbol{\alpha}_{t-1}+\eta\mathbf{e}_k) \leq \frac{1}{m}\sum_{i=1}^{m} \frac{1+y_i h_k(x_i) - r_k(x_i)}{2} \Phi\big(\mathbf{r}_{t-1}(x_i) - y_i \mathbf{h}_{t-1}(x_i) - \eta\big)$$
$$+ \frac{1}{m}\sum_{i=1}^{m} \frac{1-y_i h_k(x_i) + r_k(x_i)}{2} \Phi\big(\mathbf{r}_{t-1}(x_i) - y_i \mathbf{h}_{t-1}(x_i) + \eta\big)$$
$$+ \frac{1}{m}\sum_{i=1}^{m} \frac{-b\, r_k(x_i)}{2} c\, \Phi\big(-b\,\mathbf{r}_{t-1}(x_i) + \eta\big)$$
$$+ \frac{1}{m}\sum_{i=1}^{m} \frac{b\, r_k(x_i)}{2} c\, \Phi\big(-b\,\mathbf{r}_{t-1}(x_i) - \eta\big) + \sum_{j=1}^{N} \alpha_{t-1}\beta + \beta\eta$$

We define $J(\eta)$ to be the right-hand side of the inequality above. We will select $\eta$ as the solution of $\min_{\eta+\alpha_{t-1,k}\geq 0} J(\eta)$, which is a convex optimization problem since $J$ is convex.

### D.2.1 Exponential loss

When $\Phi(u) = \exp(u)$, the $J$ function is given by

$$J(\eta) = \frac{1}{m}\sum_{i=1}^{m} \frac{1+y_i h_k(x_i) - r_k(x_i)}{2} e^{\mathbf{r}_{t-1}(x_i) - y_i \mathbf{h}_{t-1}(x_i)} e^{-\eta}$$
$$+ \frac{1}{m}\sum_{i=1}^{m} \frac{1-y_i h_k(x_i) + r_k(x_i)}{2} e^{\mathbf{r}_{t-1}(x_i) - y_i \mathbf{h}_{t-1}(x_i)} e^{\eta}$$
$$+ \frac{1}{m}\sum_{i=1}^{m} \frac{-b\, r_k(x_i)}{2} c e^{-b\,\mathbf{r}_{t-1}(x_i)} e^{\eta}$$
$$+ \frac{1}{m}\sum_{i=1}^{m} \frac{b\, r_k(x_i)}{2} c e^{-b\,\mathbf{r}_{t-1}(x_i)} e^{-\eta} + \sum_{j=1}^{N} \alpha_{t-1}\beta + \beta\eta.$$

Since $e^{\mathbf{r}_{t-1}(x_i)-y_i\mathbf{h}_{t-1}(x_i)} = \Phi'\big(\mathbf{r}_{t-1}(x_i) - y_i\mathbf{h}_{t-1}(x_i)\big) = Z_tD_t(i,1)$ and $e^{-b\,\mathbf{r}_{t-1}(x_i)} = \Phi'\big(-b\,\mathbf{r}_{t-1}(x_i)\big) = Z_tD_t(i,2)$, it implies that

$$J(\eta) = \frac{Z_t}{m}\Big((1 - \epsilon_{t,k} - \frac{\overline{r}_{k,1}}{2})Z_{1,t}e^{-\eta} + (\epsilon_{t,k} + \frac{\overline{r}_{k,1}}{2})Z_{1,t}e^{\eta}$$
$$+ \frac{-b\,\overline{r}_{k,2}}{2}cZ_{2,t}e^{\eta} + \frac{b\,\overline{r}_{k,2}}{2}cZ_{2,t}e^{-\eta}\Big) + \sum_{j=1}^{N}\alpha_{t-1}\beta + \beta\eta.$$

For simplicity below, we define $A = Z_{1,t}(1 - \epsilon_{t,k} - \frac{\overline{r}_{k,1}}{2}) + cZ_{2,t}\frac{b\,\overline{r}_{k,2}}{2}$ and $Z = Z_{1,t}(\epsilon_{t,k} + \frac{\overline{r}_{k,1}}{2}) + cZ_{2,t}\frac{-b\,\overline{r}_{k,2}}{2}$ so that $J$ can be written as

$$J(\eta) = \frac{Z_t}{m}\Big(Ae^{-\eta} + Ze^{\eta}\Big) + \sum_{j=1}^{N}\alpha_{t-1}\beta + \beta\eta.$$

Introducing a Lagrange variable $\lambda \geq 0$, the optimization problem then becomes

$$L(\eta, \lambda) = J(\eta) - \lambda(\eta + \alpha_{t-1,k}) \text{ with } \nabla_\eta L(\eta, \lambda) = J'(\eta) - \lambda.$$

By the KKT conditions, at the solution $(\eta^*, \lambda^*)$, $J'(\eta^*) = \lambda^*$ and $\lambda^*(\eta^* + \alpha_{t-1,k}) = 0$. Thus, we can fall in one of the two following cases:

1. $(\lambda^* > 0) \Leftrightarrow (J'(\eta^*) > 0)$ and $\eta^* = -\alpha_{t-1,k}$
2. $\lambda^* = 0$ and $\eta^*$ is a solution of the equation $J(\eta^*) = 0$

The first case can be written as

$$\frac{Z_t}{m}\Big(-Ae^{\alpha_{t-1,k}} + Ze^{-\alpha_{t-1,k}}\Big) + \beta > 0 \Leftrightarrow Ae^{\alpha_{t-1,k}} - Ze^{-\alpha_{t-1,k}} < \frac{m}{Z_t}\beta.$$

For the second case we have to solve $J'(\eta) = 0$ which can be written as $e^{2\eta} + \frac{m\beta}{Z_tZ}e^{\eta} - \frac{A}{Z}$. The solution is given by

$$e^\eta = -\frac{m\beta}{2Z_tZ} + \sqrt{\Big(\frac{m\beta}{2Z_tZ}\Big)^2 + \frac{A}{Z}} \Leftrightarrow \eta = \log\Big[-\frac{m\beta}{2Z_tZ} + \sqrt{\Big(\frac{m\beta}{2Z_tZ}\Big)^2 + \frac{A}{Z}}\Big].$$

Noting that $A = Z_{1,t} - Z$, the above can be simplified to

$$\eta = \log\Big[-\frac{m\beta}{2Z_tZ} + \sqrt{\Big(\frac{m\beta}{2Z_tZ}\Big)^2 + \frac{Z_{1,t}}{Z} - 1}\Big]. \tag{11}$$

### D.2.2 Logistic loss

For the logistic loss, we have that for any $u \in \mathbb{R}$, $\Phi(-u) = \log_2(1 + e^{-u})$ and $\Phi'(-u) = \frac{1}{\log 2(1+e^u)}$. We have the following upper bound

$$\Phi(-u-v) - \Phi(-u) = \log_2\Big(\frac{1 + e^{-u} + e^{-u-v} - e^{-u}}{1 + e^{-u}}\Big) = \log_2\Big(1 + \frac{e^{-v} - 1}{e^{-u} + 1}\Big)$$
$$\leq \frac{e^{-v} - 1}{\log 2(1 + e^u)} = \Phi'(-u)(e^{-v} - 1),$$

which allows us to write

$$F(\boldsymbol{\alpha}_{t-1} + \eta\mathbf{e}_k) - F(\boldsymbol{\alpha}_{t-1}) \leq \frac{1}{m}\sum_{i=1}^{m}\Phi'(\mathbf{r}_{t-1}(x_i) - y_i\mathbf{h}_{t-1}(x_i))(e^{-\eta y_i h_k(x_i) + \eta r_k(x_i)} - 1)$$
$$+ c\,\Phi'(-b\,\mathbf{r}_{t-1}(x_i))(e^{-b\,\eta r_k(x_i)} - 1) + \beta\eta.$$

From here, we can use a very similar reasoning as the exponential loss which results in a similar expression for the step size.

# E  Convergence analysis of algorithm

In the section, we prove the convergence of the projected CD algorithm for $F(\boldsymbol{\alpha}) = \frac{1}{m}\sum_{i=1}^{m}\Phi\big(\mathbf{r}_t(x_i) - y_i\mathbf{h}_t(x_i)\big) + c\,\Phi\big(-b\,\mathbf{r}_t(x_i)\big) + \beta\sum_{j=1}^{N}\alpha_j$.

**Theorem 4.** *Assume that $\Phi$ is twice differentiable and that $\Phi''(u) > 0$ for all $u \in \mathbb{R}$. Then, the projected CD algorithm applied to $F$ converges to the solution $\boldsymbol{\alpha}^*$ of the optimization problem $\max_{\boldsymbol{\alpha}\geq 0} F(\boldsymbol{\alpha})$. If additionally $\Phi$ is strongly convex over the path of the iterates $\boldsymbol{\alpha}_t$ then there exists $\tau > 0$ and $\nu > 0$ such that for all $t > \tau$,*

$$F(\boldsymbol{\alpha}_{t+1}) - F(\boldsymbol{\alpha}^*) \leq (1 - \frac{1}{\nu})(F(\boldsymbol{\alpha}_t) - F(\boldsymbol{\alpha}^*)). \tag{12}$$

*Proof.* Let $\mathbf{H}$ be the matrix in $\mathbb{R}^{2m\times N}$ defined by $\mathbf{H}_{(i,1),j} = y_ih_j(x_i) - r_j(x_i)$ and $\mathbf{H}_{(i,2),j} = b\,r_j(x_i)$ for all $i \in [1,m]$ and for all $j \in [1,N]$, and let $\mathbf{e}_{(i,1)}$ and $\mathbf{e}_{(i,2)}$ be unit vectors in $\mathbb{R}^{2m}$. Then for any $\boldsymbol{\alpha}$, we have that $\mathbf{e}_{i,1}^T\mathbf{H}\boldsymbol{\alpha} = \sum_{j=1}^{N}\alpha_j(y_ih_j(x_i) - r_j(x_i))$ and $\mathbf{e}_{i,2}^T\mathbf{H}\boldsymbol{\alpha} = b\sum_{j=1}^{N}\alpha_j r_j(x_i)$. Thus, we can write for any $\boldsymbol{\alpha} \in \mathbb{R}^N$,

$$F(\boldsymbol{\alpha}) = G(\mathbf{H}\boldsymbol{\alpha}) + \Lambda^T\boldsymbol{\alpha}, \tag{13}$$

where $\Lambda = (\Lambda_1, \ldots, \Lambda_N)^T$ and where $G$ is the function defined by

$$G(\mathbf{u}) = \frac{1}{m}\sum_{i=1}^{m}\Phi(-e_{i,1}^T\mathbf{u}) + c\Phi(-e_{i,2}^T\mathbf{u}) = \frac{1}{m}\sum_{i=1}^{m}\Phi(-\mathbf{u}_{i,1}) + c\Phi(-\mathbf{u}_{i,2}) \tag{14}$$

for all $\mathbf{u} \in \mathbb{R}^{2m}$ with $u_{i,1}$ its $(i,1)$th coordinate and $u_{i,2}$ its $(i,2)$th coordinate. Since $\Phi$ is differentiable, the function $G$ is differentiable and $\nabla^2 G(\mathbf{u})$ is a diagonal matrix with diagonal entries $\frac{1}{m}\Phi''(-u_{i,1}) > 0$ or $\frac{c}{m}\Phi''(-u_{i,2}) > 0$ for all $i \in [1,m]$. Thus, $\nabla^2 G(\mathbf{H}\boldsymbol{\alpha})$ is positive definite for all $\boldsymbol{\alpha}$. The conditions of Theorem 2.1 of [20] are therefore satisfied for the optimization problem

$$\min_{\boldsymbol{\alpha}\geq 0} G(\mathbf{H}\boldsymbol{\alpha}) + \Lambda^T\boldsymbol{\alpha}, \tag{15}$$

thereby guaranteeing the convergence of the projected CD method applied to $F$. If additionally F is strongly convex over the sequence of $\boldsymbol{\alpha}_t$s, the by the result of [20][page 26], the Inequality 12 holds for the projected coordinate method that we are using which selects the best direction at each round, as with the Gauss-Southwell method. $\square$

# F  Calibration

In this section, we show that $L_{\mathrm{SB}}(h, r, x, y) = e^{a\,(r(x) - yh(x))} + ce^{-b\,r(x)}$ is a calibrated loss whenever $\frac{b}{a} = 2\sqrt{\frac{1-c}{c}}$. Below, let $L := L_{\mathrm{SB}}(h, r, x, y)$ and define $\eta(x) = \mathbb{P}(Y = +1|X = x)$.

**Theorem 1.** *For $a > 0$ and $b > 0$, the $\inf_{(h,r)}\mathbb{E}_{(x,y)}[L(h, r, x, y)]$ is attained at $(h_L^*, r_L^*)$ such that $\mathrm{sign}(h^*) = \mathrm{sign}(h_L^*)$ and $\mathrm{sign}(r^*) = \mathrm{sign}(r_L^*)$ if and only if $\frac{b}{a} = 2\sqrt{\frac{1-c}{c}}$.*

*Proof.* Conditioning on the label $y$, we can write the generalization error for the $L(h, r, x, y)$ as follows

$$\mathbb{E}_{(x,y)}[L(h, r, x, y)] = \mathbb{E}_{x}[\eta(x)\Psi(-h(x), r(x)) + (1 - \eta(x))\Psi(h(x), r(x))],$$

where $\Psi(-h(x), r(x)) = e^{a\,(r(x) - h(x))} + ce^{-b\,r(x)}$. For simplicity, we also let $L_{\Psi}(h(x), r(x)) = \eta(x)\Psi(-h(x), r(x)) + (1 - \eta(x))\Psi(h(x), r(x))$. Since the infimum is over all measurable functions $(h(x), r(x))$, we have that $\inf_{(h,r)}\mathbb{E}_x L_{\Psi}(h(x), r(x)) = \mathbb{E}_x \inf_{(h(x),r(x))} L_{\Psi}(h(x), r(x))$. Thus, we need to find the optimal $(u, v)$ for a fixed $x$ that minimizes $L_{\Psi}(u, v)$ over all measurable functions, which is a convex optimization problem. When $\eta(x) = 0$, the sign of the minimizers of $L_{\Psi}(u, v)$ are $u^* < 0$ and $v^* > 0$ while when $\eta(x) = 1$, the the sign of the minimizers are $u^* > 0$ and $v^* > 0$, which matches the sign of $h^*$ and $r^*$ in both cases respectively. Now for $\eta(x) \in ]0, 1[$, we take the derivative of $L_{\Psi}(u, v)$ with respect to $u$

$$\frac{\partial L_{\Psi}(u,v)}{\partial u} = -\eta(x)a\,e^{a\,(v-u)} + (1 - \eta(x))a\,e^{a\,(u+v)}.$$

Setting it to zero and solving for $u$, we have that $u^* = \frac{1}{2a}\log(\frac{\eta(x)}{1-\eta(x)})$. We can now see that $u^* > 0$ if $\eta(x) > \frac{1}{2}$ and $u^* \leq 0$ if $\eta(x) \leq \frac{1}{2}$. Recalling that $h^* = \eta(x) - \frac{1}{2}$, we can conclude that the sign of $u^*$ matches the sign of $h^*$.

We now take the derivative of $L_\Psi(u^*, v)$ with respect to $v$

$$\frac{\partial L_\Psi(u^*,v)}{\partial v} = \eta(x)e^{a(v-u^*)} + (1-\eta(x))e^{a(v+u^*)} + c(-b)e^{-bv}.$$

Setting it equal to zero and using the fact that $\eta(x)e^{-au^*} + (1-\eta(x))e^{au^*} = 2\sqrt{\eta(x)(1-\eta(x))}$, we have that

$$v^* = \frac{1}{a+b}\log\left(\frac{cb}{2a}\sqrt{\frac{1}{\eta(x)(1-\eta(x))}}\right).$$

Now, we know that the Bayes classifiers $(h^*, r^*)$ satisfy $h^* = \eta(x) - \frac{1}{2}$ and $r^* = |h^*| - \frac{1}{2} + c$ so that the following holds

$$\eta(x)(1-\eta(x)) = \frac{1}{4} - (h^*)^2 = \frac{1}{4} - (r^* + \frac{1}{2} - c)^2.$$

Thus, we can replace $\eta(x)(1-\eta(x))$ in the definition of $v^*$ to arrive at this equation

$$v^* = \frac{1}{a+b}\log\left(\frac{cb}{2a}\sqrt{\frac{1}{\frac{1}{4}-(r^*+\frac{1}{2}-c)^2}}\right).$$

We now analyze when $v^* > 0$ which is equivalent to

$$\frac{1}{a+b}\log\left(\frac{cb}{2a}\sqrt{\frac{1}{\frac{1}{4}-(r^*+\frac{1}{2}-c)^2}}\right) > 0 \Leftrightarrow \frac{cb}{2a}\sqrt{\frac{1}{\frac{1}{4}-(r^*+\frac{1}{2}-c)^2}} > 1$$

$$\Leftrightarrow \frac{cb}{2a} > \sqrt{\frac{1}{4} - (r^* + \frac{1}{2} - c)^2}.$$

Since $\sqrt{\frac{1}{4} - (\frac{1}{2} - c)^2} > \sqrt{\frac{1}{4} - (r^* + \frac{1}{2} - c)^2}$ for $r^* > 0$ and using the fact that $c(1-c) = \frac{1}{4} - (\frac{1}{2} - c)^2$, we need that $\frac{cb}{2a} \geq \sqrt{c(1-c)}$. By similar reasoning for $v^* \leq 0$, we need that $\frac{cb}{2a} \leq \sqrt{c(1-c)}$. Thus, we can conclude that the sign of $v^*$ matches the sign of $r^*$ if and only if $\frac{cb}{2a} = \sqrt{c(1-c)}$. $\qquad\square$

# G  Abstention stumps

Under the assumptions of Section 4.3, the derivative of $F$ can be simplified as follows

$$F'(\boldsymbol{\alpha}_{t-1}, \mathbf{e}_j) = \frac{Z_t}{m}\left(-\sum_{i:y_i h_j(x_i)=+1} D_t(i,1) + \sum_{i:y_i h_j(x_i)=-1} D_t(i,1) + \sum_{i:r_j(x_i)=1} D_t(i,1)\right.$$
$$\left. - cb\sum_{i:r_j(x_i)=1} D_t(i,2)\right) + \beta \tag{16}$$

From the definition of $D(i,1)$ and the assumptions on $h(x)$ and $r(x)$, the following holds

$$\sum_{i=1}^{m} D_t(i,1) = \sum_{i:y_i h_j(x_i)=+1} D_t(i,1) + \sum_{i:y_i h_j(x_i)=-1} D_t(i,1) + \sum_{i:r_j(x_i)=1} D_t(i,1)$$

Solving for $\sum_{i:y_i h_j(x_i)=+1} D_t(i,1)$ and plugging it in Equation 16, we can simplify the derivative

$$F'(\boldsymbol{\alpha}_{t-1}, \mathbf{e}_j) = \frac{Z_t}{m}\left(2\sum_{i:y_i h_j(x_i)=-1} D_t(i,1) + 2\sum_{i:r_j(x_i)=1} D_t(i,1) - \sum_{i=1}^{m} D_t(i,1) - cb\sum_{i:r_j(x_i)=1} D_t(i,2)\right) + \beta$$

$$= \frac{Z_t}{m}\left(2Z_{1,t}\epsilon_{t,j} + 2Z_{1,t}\bar{r}_{j,1} - cb\, Z_{2,t}\bar{r}_{j,2} - Z_{1,t}\right) + \beta$$

Thus, the optimal descent direction is $k = \text{argmin}_{j\in[1,N]} 2Z_{1,t}\epsilon_{t,j} + 2Z_{1,t}\bar{r}_{j,1} - cb\, Z_{2,t}\bar{r}_{j,2}$

Below, we provide the proof of the lemma that was needed to decouple the optimization problem for the abstention stumps.

**Lemma 5.** *The optimization problem without the constraint* $(\theta_1 < \theta_2)$ *can be decomposed as follows:*

$$\operatorname*{argmin}_{\theta_1,\theta_2}\sum_{i=1}^{m}2D_t(i,1)[1_{y_i=+1}1_{X_i\leq\theta_1}+1_{y_i=-1}1_{X_i>\theta_2}]+\big(2D_t(i,1)-cb\,D_t(i,2)\big)1_{\theta_1<X_i\leq\theta_2}.$$

$$=\operatorname*{argmin}_{\theta_1}\sum_{i=1}^{m}2D_t(i,1)1_{y_i=+1}1_{X_i\leq\theta_1}+\big(2D_t(i,1)-cb\,D_t(i,2)\big)1_{\theta_1<X_i}$$

$$+\operatorname*{argmin}_{\theta_2}\sum_{i=1}^{m}2D_t(i,1)1_{y_i=-1}1_{X_i>\theta_2}+\big(2D_t(i,1)-cb\,D_t(i,2)\big)1_{X_i\leq\theta_2}.$$

*Proof.* For simplicity below, let $\kappa=\big(2D_t(i,1)-cb\,D_t(i,2)\big)$ and observe that the following identity holds:

$$1_{\theta_1<X_i\leq\theta_2}=1_{\theta_1<X_i}+1_{X_i\leq\theta_2}-1$$

In view of that, we can write

$$\operatorname*{argmin}_{\theta_1,\theta_2}\sum_{i=1}^{m}2D_t(i,1)[1_{y_i=+1}1_{X_i\leq\theta_1}+1_{y_i=-1}1_{X_i>\theta_2}]+\kappa 1_{\theta_1<X_i\leq\theta_2}$$

$$=\operatorname*{argmin}_{\theta_1,\theta_2}\sum_{i=1}^{m}2D_t(i,1)[1_{y_i=+1}1_{X_i\leq\theta_1}+1_{y_i=-1}1_{X_i>\theta_2}]+\kappa[1_{\theta_1<X_i}+1_{X_i\leq\theta_2}-1]$$

$$=\operatorname*{argmin}_{\theta_1,\theta_2}\sum_{i=1}^{m}2D_t(i,1)[1_{y_i=+1}1_{X_i\leq\theta_1}+1_{y_i=-1}1_{X_i>\theta_2}]+\kappa 1_{\theta_1<X_i}$$
$$+\kappa 1_{X_i\leq\theta_2}-\kappa$$

$$=\operatorname*{argmin}_{\theta_1,\theta_2}\sum_{i=1}^{m}2D_t(i,1)[1_{y_i=+1}1_{X_i\leq\theta_1}+1_{y_i=-1}1_{X_i>\theta_2}]+\kappa 1_{\theta_1<X_i}$$
$$+\kappa 1_{X_i\leq\theta_2}$$

$$=\operatorname*{argmin}_{\theta_1}\sum_{i=1}^{m}2D_t(i,1)1_{y_i=+1}1_{X_i\leq\theta_1}+\kappa 1_{\theta_1<X_i}$$

$$+\operatorname*{argmin}_{\theta_2}\sum_{i=1}^{m}2D_t(i,1)1_{y_i=-1}1_{X_i>\theta_2}+\kappa 1_{X_i\leq\theta_2}.$$

$\square$

## H  Alternative surrogate, $L_{\mathrm{MB}}$

In this section, we derive the boosting algorithm for the surrogate loss

$$L_{\mathrm{MB}}(\mathbf{h},\mathbf{r},x,y)=\max\Big(\Phi_1\big(a\,[\mathbf{r}(x)-y\mathbf{h}(x)]\big),c\,\Phi_2\big(-b\,\mathbf{r}(x)\big)\Big). \tag{17}$$

By a similar reasoning as Section 4, the objective function $F(\boldsymbol{\alpha})$ of our optimization problem is given by the following

$$F(\boldsymbol{\alpha})=\frac{1}{m}\sum_{i=1}^{m}\max\Big(e^{a\,[\mathbf{r}_t(x_i)-y_i\mathbf{h}_t(x_i)]},cb\,e^{-b\,\mathbf{r}_t(x_i)}\Big)+\beta\sum_{j=1}^{N}\alpha_j.$$

For simplicity, we define $u_t(i)=e^{a\,[\mathbf{r}_t(x_i)-y_i\mathbf{h}_t(x_i)]}$, $v_t(i)=cb\,e^{-b\,\mathbf{r}_t(x_i)}$ and $w_t(i)=\max(u_t(i),v_t(i))$. We also let $1_{u_t(i)}$ be the indicator functions that equals 1 if $u_t(i)\geq v_t(i)$ and similarly $1_{v_t(i)}$ be the indicator functions that equals 1 if $v_t(i)>u_t(i)$. For any $t\in[1,T]$, we also define the distribution

$$\mathcal{D}_t(i)=\frac{w_{t-1}(i)}{Z_t}, \tag{18}$$

where $Z_t$ is the normalization factor given by $Z_t = \sum_{i=1}^{m} w_{t-1}(i)$.

We then apply projected coordinate descent to this objective function. Notice that our objective $F$ is differentiable everywhere except when $u_t(i) = v_t(i)$. A true maximum descent algorithm would choose the element of the subgradient that is closest to $0$ as the descent direction. However, since this event is rare in our case, we arbitrarily a pick descent direction that is an element of the subgradient. For simplicity below, we will use the symbol $F'(\boldsymbol{\alpha}_{t-1}, \mathbf{e}_j)$ to denote the directional derivative with the added condition that for the non-differentiable point, we choose the direction that is an element of the subgradient.

### H.0.3   Direction and step

At each iteration $t-1$, the direction $\mathbf{e}_k$ selected by projected CD is $k = \operatorname{argmax}_{j \in [1,N]} |F'(\boldsymbol{\alpha}_{t-1}, \mathbf{e}_j)|$ where

$$
\begin{aligned}
&F'(\boldsymbol{\alpha}_{t-1}, \mathbf{e}_j) \\
&= \frac{1}{m} \sum_{i=1}^{m} \Big( a\,[-y_i h_j(x_i) + r_j(x_i)] 1_{u_{t-1}(i)} - cb\, r_j(x_i) 1_{v_{t-1}(i)} \Big) w_{t-1}(i) + \beta \\
&= \frac{1}{m} \sum_{i=1}^{m} \Big( -a\, y_i h_j(x_i) 1_{u_{t-1}(i)} - [-(a + cb) 1_{u_{t-1}(i)} + cb] r_j(x_i) \Big) w_{t-1}(i) + \beta \\
&= \frac{1}{m} \sum_{i=1}^{m} \Big( -a\, y_i h_j(x_i) 1_{u_{t-1}(i)} - [-(a + cb) 1_{u_{t-1}(i)} + cb] r_j(x_i) \Big) \mathcal{D}_t(i) Z_t + \beta.
\end{aligned} \tag{19}
$$

The step can simply be found via line search or other numerical methods.

### H.1   Abstention stumps

We focus in on a special case where the base classifiers have a specific form defined as follows: $h(x)$ takes values in $\{-1, 0, 1\}$ and $r(x)$ take values in $\{0, 1\}$. We also have the added the condition that for each sample point $x$, only one of the two components of $(h(x), r(x))$ is non-zero. Under this setting, Equation 19 can be simplified as follows. The derivative of $F$ is given by

$$
F'(\boldsymbol{\alpha}_{t-1}, \mathbf{e}_j) = \frac{1}{m} \sum_{i=1}^{m} \Big( a\,[-y_i h_j(x_i) + r_j(x_i)] 1_{u_{t-1}(i)} - cb\, r_j(x_i) 1_{v_{t-1}(i)} \Big) D_t(i) Z_t + \beta,
$$

which can be rewritten as

$$
\begin{aligned}
&= \frac{Z_t}{m} \Big( a\,\big[ \sum_{i:y_i h_j(x_i)=-1} 1_{u_{t-1}(i)} D_t(i) - \sum_{i:y_i h_j(x_i)=+1} 1_{u_{t-1}(i)} D_t(i) + \sum_{i:r_j(x_i)=1} 1_{u_{t-1}(i)} D_t(i) \big] \\
&\quad - cb \sum_{i:r_j(x_i)=1} D_t(i) 1_{v_{t-1}(i)} \Big) + \beta.
\end{aligned} \tag{20}
$$

From the assumptions on $h(x)$ and $r(x)$, the relation below holds:

$$
\sum_{i=1}^{m} D_t(i) 1_{u_{t-1}(i)} = \sum_{i:y_i h_j(x_i)=+1} D_t(i) 1_{u_{t-1}(i)} + \sum_{i:y_i h_j(x_i)=-1} D_t(i) 1_{u_{t-1}(i)} + \sum_{i:r_j(x_i)=1} D_t(i) 1_{u_{t-1}(i)},
$$

which is equivalent to the following

$$
- \sum_{i:y_i h_j(x_i)=+1} D_t(i) 1_{u_{t-1}(i)} = \sum_{i:y_i h_j(x_i)=-1} D_t(i) 1_{u_{t-1}(i)} + \sum_{i:r_j(x_i)=1} D_t(i) 1_{u_{t-1}(i)} - \sum_{i=1}^{m} D_t(i) 1_{u_{t-1}(i)}.
$$

Table 1: For each data set, we report the sample size and the number of features.

| Data Sets | Sample Size | Feature |
|-----------|-------------|---------|
| australian | 690 | 14 |
| cod | 369 | 8 |
| skin | 400 | 3 |
| banknote | 1,372 | 4 |
| haberman | 306 | 3 |
| pima | 768 | 8 |

Plugging this into equation 20, we have that

$$
\begin{aligned}
& F'(\boldsymbol{\alpha}_{t-1}, \mathbf{e}_j) \\
&= \frac{Z_t}{m}\Big(a\,[2\sum_{i:y_i h_j(x_i)=-1} 1_{u_{t-1}(i)}D_t(i) + 2\sum_{i:r_j(x_i)=1} 1_{u_{t-1}(i)}D_t(i) - \sum_{i=1}^{m} 1_{u_{t-1}(i)}D_t(i)] \quad (21) \\
&\quad - cb\sum_{i:r_j(x_i)=1} D_t(i)1_{v_{t-1}(i)}\Big) + \beta.
\end{aligned}
$$

This in turn implies that our weak learning algorithm is given by the following:

$$
l(\theta_1,\theta_2) = \underset{i\sim D}{\mathbb{E}}[2a\,1_{u(i)}[1_{y_i=+1}1_{h_{\theta_1,\theta_2}(x)=-1} + 1_{y_i=-1}1_{h_{\theta_1,\theta_2}(x)=1}] + [2a\,1_{u(i)} - cb\,1_{v(i)}]\,1_{r_{\theta_1,\theta_2}(x)=1}].
$$

The following lemma allows us to decouple the optimization problem into two optimization problems with respect to $\theta_1$ and $\theta_2$ that can be solved in linear time.

**Lemma 7.** *The optimization problem without the constraint $(\theta_1 < \theta_2)$ can be decomposed as follows:*

$$
\begin{aligned}
& \underset{\theta_1,\theta_2}{\operatorname{argmin}}\,\underset{i\sim D}{\mathbb{E}}\left(2a\,1_{u(i)}[1_{y_i=+1}1_{X_i\leq\theta_1} + 1_{y_i=-1}1_{X_i>\theta_2}] + [2a\,1_{u(i)} - cb\,1_{v(i)}]1_{\theta_1<X_i\leq\theta_2}\right) \\
&= \underset{\theta_1}{\operatorname{argmin}}\,\underset{i\sim D}{\mathbb{E}}\left(2a\,1_{u(i)}1_{y_i=+1}1_{X_i\leq\theta_1} + [2a\,1_{u(i)} - cb\,1_{v(i)}]1_{\theta_1<X_i}\right) \\
&\quad + \underset{\theta_2}{\operatorname{argmin}}\,\underset{i\sim D}{\mathbb{E}}\left(2a\,1_{u(i)}1_{y_i=-1}1_{X_i>\theta_2} + [2a\,1_{u(i)} - cb\,1_{v(i)}]1_{X_i\leq\theta_2}\right).
\end{aligned}
$$

*Proof.* For simplicity, let $\kappa = 2a\,1_{u(i)} - cb\,1_{v(i)}$ and observe that the following identity holds:

$$
1_{\theta_1<X_i\leq\theta_2} = 1_{\theta_1<X_i} + 1_{X_i\leq\theta_2} - 1
$$

In view of that, we can write

$$
\begin{aligned}
& \underset{\theta_1,\theta_2}{\operatorname{argmin}}\,\underset{i\sim D}{\mathbb{E}}[2a\,1_{u(i)}[1_{y_i=+1}1_{X_i\leq\theta_1} + 1_{y_i=-1}1_{X_i>\theta_2}] + \kappa 1_{\theta_1<X_i\leq\theta_2}] \\
&= \underset{\theta_1,\theta_2}{\operatorname{argmin}}\,\underset{i\sim D}{\mathbb{E}}[2a\,1_{u(i)}[1_{y_i=+1}1_{X_i\leq\theta_1} + 1_{y_i=-1}1_{X_i>\theta_2}] + \kappa(1_{\theta_1<X_i} + 1_{X_i\leq\theta_2} - 1)] \\
&= \underset{\theta_1,\theta_2}{\operatorname{argmin}}\,\underset{i\sim D}{\mathbb{E}}[2a\,1_{u(i)}1_{y_i=+1}1_{X_i\leq\theta_1} + \kappa 1_{\theta_1<X_i} + 2a\,1_{u(i)}1_{y_i=-1}1_{X_i>\theta_2} \\
&\qquad\qquad + \kappa 1_{X_i\leq\theta_2} - \kappa] \\
&= \underset{\theta_1,\theta_2}{\operatorname{argmin}}\,\underset{i\sim D}{\mathbb{E}}[2a\,1_{u(i)}1_{y_i=+1}1_{X_i\leq\theta_1} + \kappa 1_{\theta_1<X_i}] \\
&\quad + \underset{\theta_1,\theta_2}{\operatorname{argmin}}\,\underset{i\sim D}{\mathbb{E}}[2a\,1_{u(i)}1_{y_i=-1}1_{X_i>\theta_2} + \kappa 1_{X_i\leq\theta_2}]
\end{aligned}
$$

which completes the proof. □

# I   Data sets

Table 1 shows the sample size and number of features for each data set used in our experiments.

# J Experiments

In this appendix, we report the results of several experiments by presenting different tables in order to compare the three algorithms studied in this paper: TSB, DHL, and BA. In each table, we provide the average and standard deviation on the test set for the hyper parameter configurations that admitted the smallest abstention loss on the validation set. Overall, these results reveal that BA yields a significant improvement in practice for all the data sets across different values of cost $c$.

Table 2 gives the average abstention loss on the test set for TSB, DHL, and BA algorithms. Across almost all the different values of cost $c$, the BA algorithm attains the smallest abstention loss compared with the TSB and DHL algorithms. On some datasets, the TSB performs better than the DHL algorithm, but on other datasets, its performance largely deteriorates. We also see that the effects of changing the cost $c$ of rejection for some datasets is much stronger than for other datasets. For example, the `pima` dataset has a large change in abstention loss as $c$ increases while for `banknote` dataset the difference in abstention loss is very small. These changes reflect the changes in the fraction of points rejected by the algorithms, see Table 3. Note that this effect also depends on the algorithm as seen in the `cod` dataset where BA algorithm's abstention loss changes only slightly while for the other two algorithms the difference is much higher as $c$ increases.

Table 3 shows the fraction of points that are rejected on the test set. For all three algorithms, the fraction of points rejected decreases as the cost $c$ of rejection increases. Moreover, the fraction of points rejected is much higher for some datasets. For most values of $c$, the DHL algorithm appears to reject less frequently, but its abstention loss is also higher. For `haberman`, `australian`, and `pima` datasets, the TSB algorithm rejection rates is quite high, which reinforces our claim that DHL and BA algorithms are better algorithms. Finally, Table 4, presents the classification loss on non-rejected points for different values of $c$. As $c$ increases, we see that more points are classified incorrectly, which is in accordance with the previous table since it shows that we are also rejecting less points.

Table 2: Average abstention loss along with the standard deviations on the test set for the TSB Algorithm, DHL Algorithm and BA Algorithm

| Cost | skin TSB | skin DHL | skin BA | cod TSB | cod DHL | cod BA |
|---|---|---|---|---|---|---|
| 0.05 | $0.0482 \pm 0.0156$ | $0.024 \pm 0.016$ | $0.0258 \pm 0.0157$ | $0.0384 \pm 0.00926$ | $0.044 \pm 0.034$ | $0.0386 \pm 0.0152$ |
| 0.1 | $0.059 \pm 0.0345$ | $0.061 \pm 0.031$ | $0.0373 \pm 0.0166$ | $0.0784 \pm 0.0176$ | $0.077 \pm 0.028$ | $0.0624 \pm 0.00367$ |
| 0.15 | $0.0822 \pm 0.0141$ | $0.091 \pm 0.031$ | $0.0595 \pm 0.0174$ | $0.0807 \pm 0.0138$ | $0.123 \pm 0.030$ | $0.0593 \pm 0.0279$ |
| 0.2 | $0.052 \pm 0.0262$ | $0.128 \pm 0.036$ | $0.04 \pm 0.0185$ | $0.0903 \pm 0.0245$ | $0.175 \pm 0.031$ | $0.0654 \pm 0.0274$ |
| 0.25 | $0.0667 \pm 0.0304$ | $0.158 \pm 0.041$ | $0.0425 \pm 0.0174$ | $0.0831 \pm 0.00896$ | $0.204 \pm 0.026$ | $0.0676 \pm 0.0304$ |
| 0.3 | $0.037 \pm 0.0226$ | $0.177 \pm 0.044$ | $0.0403 \pm 0.0162$ | $0.117 \pm 0.0151$ | $0.230 \pm 0.022$ | $0.0659 \pm 0.0285$ |
| 0.35 | $0.0593 \pm 0.0272$ | $0.204 \pm 0.056$ | $0.0477 \pm 0.0144$ | $0.11 \pm 0.0182$ | $0.259 \pm 0.029$ | $0.0581 \pm 0.0313$ |
| 0.4 | $0.0907 \pm 0.0125$ | $0.231 \pm 0.067$ | $0.0567 \pm 0.0181$ | $0.106 \pm 0.0271$ | $0.273 \pm 0.026$ | $0.0692 \pm 0.0372$ |
| 0.45 | $0.0693 \pm 0.033$ | $0.215 \pm 0.066$ | $0.0525 \pm 0.0186$ | $0.12 \pm 0.0246$ | $0.276 \pm 0.025$ | $0.065 \pm 0.0364$ |

| Cost | haberman TSB | haberman DHL | haberman BA | pima TSB | pima DHL | pima BA |
|---|---|---|---|---|---|---|
| 0.05 | $0.05 \pm 0.0$ | $0.050 \pm 0.000$ | $0.05 \pm 0.0$ | $0.0512 \pm 0.00247$ | $0.068 \pm 0.039$ | $0.05 \pm 0.0$ |
| 0.1 | $0.103 \pm 0.00581$ | $0.143 \pm 0.027$ | $0.1 \pm 0.0$ | $0.106 \pm 0.00787$ | $0.176 \pm 0.009$ | $0.1 \pm 0.0$ |
| 0.15 | $0.15 \pm 0.000968$ | $0.213 \pm 0.037$ | $0.173 \pm 0.0458$ | $0.146 \pm 0.00567$ | $0.218 \pm 0.023$ | $0.157 \pm 0.0221$ |
| 0.2 | $0.204 \pm 0.0101$ | $0.233 \pm 0.036$ | $0.214 \pm 0.0256$ | $0.195 \pm 0.00489$ | $0.238 \pm 0.021$ | $0.172 \pm 0.00859$ |
| 0.25 | $0.25 \pm 0.0$ | $0.256 \pm 0.027$ | $0.234 \pm 0.0238$ | $0.235 \pm 0.00669$ | $0.241 \pm 0.025$ | $0.19 \pm 0.0211$ |
| 0.3 | $0.303 \pm 0.0147$ | $0.264 \pm 0.019$ | $0.244 \pm 0.0196$ | $0.285 \pm 0.00428$ | $0.247 \pm 0.026$ | $0.201 \pm 0.0114$ |
| 0.35 | $0.34 \pm 0.0123$ | $0.261 \pm 0.024$ | $0.265 \pm 0.0325$ | $0.327 \pm 0.00874$ | $0.250 \pm 0.027$ | $0.22 \pm 0.016$ |
| 0.4 | $0.383 \pm 0.0192$ | $0.262 \pm 0.028$ | $0.272 \pm 0.033$ | $0.374 \pm 0.0079$ | $0.255 \pm 0.028$ | $0.234 \pm 0.0134$ |
| 0.45 | $0.441 \pm 0.0235$ | $0.258 \pm 0.022$ | $0.275 \pm 0.0301$ | $0.422 \pm 0.0114$ | $0.260 \pm 0.034$ | $0.249 \pm 0.0171$ |

| Cost | australian TSB | australian DHL | australian BA | banknote TSB | banknote DHL | banknote BA |
|---|---|---|---|---|---|---|
| 0.05 | $0.0499 \pm 0.000145$ | $0.112 \pm 0.033$ | $0.0564 \pm 0.0117$ | $0.000873 \pm 0.00139$ | $0.091 \pm 0.059$ | $0.00247 \pm 0.00195$ |
| 0.1 | $0.0867 \pm 0.00455$ | $0.120 \pm 0.024$ | $0.0777 \pm 0.016$ | $0.00284 \pm 0.00299$ | $0.082 \pm 0.070$ | $0.00705 \pm 0.00632$ |
| 0.15 | $0.13 \pm 0.00615$ | $0.128 \pm 0.025$ | $0.093 \pm 0.0155$ | $0.00411 \pm 0.00108$ | $0.081 \pm 0.076$ | $0.0044 \pm 0.00411$ |
| 0.2 | $0.168 \pm 0.00612$ | $0.130 \pm 0.036$ | $0.111 \pm 0.0215$ | $0.00131 \pm 0.00197$ | $0.049 \pm 0.020$ | $0.00611 \pm 0.00509$ |
| 0.25 | $0.209 \pm 0.00898$ | $0.134 \pm 0.038$ | $0.12 \pm 0.0171$ | $0.000727 \pm 0.00068$ | $0.061 \pm 0.022$ | $0.00636 \pm 0.00381$ |
| 0.3 | $0.244 \pm 0.0126$ | $0.137 \pm 0.038$ | $0.137 \pm 0.0252$ | $0.00371 \pm 0.00239$ | $0.083 \pm 0.025$ | $0.00735 \pm 0.00397$ |
| 0.35 | $0.294 \pm 0.0141$ | $0.141 \pm 0.039$ | $0.144 \pm 0.0263$ | $0.0096 \pm 0.00426$ | $0.087 \pm 0.052$ | $0.00833 \pm 0.00465$ |
| 0.4 | $0.335 \pm 0.0254$ | $0.148 \pm 0.042$ | $0.151 \pm 0.0273$ | $0.00422 \pm 0.00281$ | $0.119 \pm 0.028$ | $0.00785 \pm 0.00492$ |
| 0.45 | $0.365 \pm 0.0223$ | $0.150 \pm 0.046$ | $0.145 \pm 0.0337$ | $0.00447 \pm 0.0038$ | $0.136 \pm 0.027$ | $0.00738 \pm 0.00399$ |

Table 3: Average fraction of points rejected along with the standard deviations on the test set for the TSB Algorithm, DHL Algorithm and BA Algorithm

| Cost | skin TSB | skin DHL | skin BA | cod TSB | cod DHL | cod BA |
|------|----------|----------|---------|---------|---------|--------|
| 0.05 | $0.497 \pm 0.207$ | $0.180 \pm 0.044$ | $0.317 \pm 0.111$ | $0.605 \pm 0.0523$ | $0.170 \pm 0.045$ | $0.557 \pm 0.0972$ |
| 0.1 | $0.0567 \pm 0.0226$ | $0.158 \pm 0.047$ | $0.173 \pm 0.0374$ | $0.189 \pm 0.074$ | $0.146 \pm 0.049$ | $0.543 \pm 0.0785$ |
| 0.15 | $0.237 \pm 0.13$ | $0.125 \pm 0.032$ | $0.13 \pm 0.0476$ | $0.376 \pm 0.0563$ | $0.132 \pm 0.039$ | $0.197 \pm 0.0488$ |
| 0.2 | $0.0267 \pm 0.0271$ | $0.100 \pm 0.025$ | $0.0667 \pm 0.0279$ | $0.168 \pm 0.0236$ | $0.065 \pm 0.026$ | $0.0568 \pm 0.0447$ |
| 0.25 | $0.0267 \pm 0.0271$ | $0.092 \pm 0.033$ | $0.0633 \pm 0.0287$ | $0.127 \pm 0.0202$ | $0.038 \pm 0.018$ | $0.0432 \pm 0.0313$ |
| 0.3 | $0.0233 \pm 0.0309$ | $0.090 \pm 0.051$ | $0.0567 \pm 0.0226$ | $0.146 \pm 0.0335$ | $0.027 \pm 0.021$ | $0.0486 \pm 0.0303$ |
| 0.35 | $0.0267 \pm 0.0309$ | $0.075 \pm 0.051$ | $0.06 \pm 0.0309$ | $0.168 \pm 0.0303$ | $0.014 \pm 0.010$ | $0.027 \pm 0.0242$ |
| 0.4 | $0.06 \pm 0.0501$ | $0.032 \pm 0.011$ | $0.05 \pm 0.035$ | $0.151 \pm 0.0405$ | $0.000 \pm 0.000$ | $0.0378 \pm 0.0262$ |
| 0.45 | $0.08 \pm 0.0323$ | $0.005 \pm 0.007$ | $0.05 \pm 0.035$ | $0.159 \pm 0.0563$ | $0.000 \pm 0.000$ | $0.0243 \pm 0.0262$ |

| Cost | haberman TSB | haberman DHL | haberman BA | pima TSB | pima DHL | pima BA |
|------|--------------|--------------|-------------|----------|----------|---------|
| 0.05 | $1.0 \pm 0.0$ | $1.000 \pm 0.000$ | $1.0 \pm 0.0$ | $0.999 \pm 0.0026$ | $0.884 \pm 0.258$ | $1.0 \pm 0.0$ |
| 0.1 | $0.997 \pm 0.00645$ | $0.738 \pm 0.183$ | $1.0 \pm 0.0$ | $0.927 \pm 0.0311$ | $0.304 \pm 0.072$ | $1.0 \pm 0.0$ |
| 0.15 | $0.997 \pm 0.00645$ | $0.348 \pm 0.123$ | $0.852 \pm 0.297$ | $0.925 \pm 0.0408$ | $0.143 \pm 0.031$ | $0.321 \pm 0.0364$ |
| 0.2 | $0.939 \pm 0.0313$ | $0.148 \pm 0.053$ | $0.216 \pm 0.0546$ | $0.901 \pm 0.043$ | $0.078 \pm 0.024$ | $0.33 \pm 0.0405$ |
| 0.25 | $1.0 \pm 0.0$ | $0.039 \pm 0.015$ | $0.187 \pm 0.0582$ | $0.894 \pm 0.038$ | $0.055 \pm 0.007$ | $0.249 \pm 0.0359$ |
| 0.3 | $0.935 \pm 0.0177$ | $0.016 \pm 0.028$ | $0.255 \pm 0.131$ | $0.923 \pm 0.0258$ | $0.039 \pm 0.015$ | $0.262 \pm 0.0343$ |
| 0.35 | $0.945 \pm 0.0718$ | $0.007 \pm 0.015$ | $0.0581 \pm 0.06$ | $0.93 \pm 0.023$ | $0.038 \pm 0.024$ | $0.238 \pm 0.0416$ |
| 0.4 | $0.9 \pm 0.0664$ | $0.007 \pm 0.009$ | $0.0516 \pm 0.064$ | $0.918 \pm 0.0292$ | $0.034 \pm 0.014$ | $0.23 \pm 0.0404$ |
| 0.45 | $0.923 \pm 0.0373$ | $0.013 \pm 0.014$ | $0.0516 \pm 0.064$ | $0.919 \pm 0.0369$ | $0.026 \pm 0.009$ | $0.219 \pm 0.0414$ |

| Cost | australian TSB | australian DHL | australian BA | banknote TSB | banknote DHL | banknote BA |
|------|----------------|----------------|---------------|--------------|--------------|-------------|
| 0.05 | $0.999 \pm 0.0029$ | $0.151 \pm 0.037$ | $0.346 \pm 0.0416$ | $0.00291 \pm 0.00356$ | $0.799 \pm 0.119$ | $0.00582 \pm 0.00493$ |
| 0.1 | $0.809 \pm 0.0566$ | $0.068 \pm 0.017$ | $0.328 \pm 0.0452$ | $0.00655 \pm 0.00424$ | $0.075 \pm 0.021$ | $0.00509 \pm 0.00291$ |
| 0.15 | $0.772 \pm 0.0745$ | $0.049 \pm 0.019$ | $0.262 \pm 0.0436$ | $0.0225 \pm 0.00842$ | $0.060 \pm 0.006$ | $0.00509 \pm 0.00291$ |
| 0.2 | $0.765 \pm 0.0597$ | $0.036 \pm 0.013$ | $0.177 \pm 0.027$ | $0.00291 \pm 0.00356$ | $0.072 \pm 0.014$ | $0.00509 \pm 0.00291$ |
| 0.25 | $0.794 \pm 0.0245$ | $0.030 \pm 0.006$ | $0.142 \pm 0.046$ | $0.00291 \pm 0.00272$ | $0.066 \pm 0.016$ | $0.00509 \pm 0.00291$ |
| 0.3 | $0.793 \pm 0.0469$ | $0.030 \pm 0.008$ | $0.142 \pm 0.0254$ | $0.00509 \pm 0.00291$ | $0.058 \pm 0.017$ | $0.00509 \pm 0.00291$ |
| 0.35 | $0.816 \pm 0.036$ | $0.025 \pm 0.011$ | $0.114 \pm 0.0312$ | $0.0233 \pm 0.0109$ | $0.041 \pm 0.025$ | $0.00509 \pm 0.00291$ |
| 0.4 | $0.801 \pm 0.0899$ | $0.010 \pm 0.006$ | $0.0145 \pm 0.0152$ | $0.00509 \pm 0.00436$ | $0.048 \pm 0.005$ | $0.00509 \pm 0.00291$ |
| 0.45 | $0.801 \pm 0.0551$ | $0.004 \pm 0.006$ | $0.0812 \pm 0.0288$ | $0.00509 \pm 0.00291$ | $0.052 \pm 0.012$ | $0.00509 \pm 0.00291$ |

Table 4: Average classification error on non-rejected points along with the standard deviation for the TSB Algorithm, DHL Algorithm and BA Algorithm

| Cost | skin TSB | skin DHL | skin BA | cod TSB | cod DHL | cod BA |
|---|---|---|---|---|---|---|
| 0.05 | $0.0233 \pm 0.0249$ | $0.015 \pm 0.016$ | $0.01 \pm 0.0133$ | $0.00811 \pm 0.0108$ | $0.035 \pm 0.034$ | $0.0108 \pm 0.0132$ |
| 0.1 | $0.0533 \pm 0.0356$ | $0.045 \pm 0.033$ | $0.02 \pm 0.0163$ | $0.0595 \pm 0.0202$ | $0.062 \pm 0.030$ | $0.00811 \pm 0.00662$ |
| 0.15 | $0.0467 \pm 0.0306$ | $0.073 \pm 0.031$ | $0.04 \pm 0.0133$ | $0.0243 \pm 0.0132$ | $0.103 \pm 0.031$ | $0.0297 \pm 0.0216$ |
| 0.2 | $0.0467 \pm 0.0245$ | $0.108 \pm 0.034$ | $0.0267 \pm 0.0226$ | $0.0568 \pm 0.0248$ | $0.162 \pm 0.030$ | $0.0541 \pm 0.0256$ |
| 0.25 | $0.06 \pm 0.0327$ | $0.135 \pm 0.037$ | $0.0267 \pm 0.0226$ | $0.0514 \pm 0.0101$ | $0.195 \pm 0.026$ | $0.0568 \pm 0.0262$ |
| 0.3 | $0.03 \pm 0.0194$ | $0.150 \pm 0.035$ | $0.0233 \pm 0.0226$ | $0.073 \pm 0.0108$ | $0.222 \pm 0.023$ | $0.0514 \pm 0.0232$ |
| 0.35 | $0.05 \pm 0.0279$ | $0.178 \pm 0.045$ | $0.0267 \pm 0.0226$ | $0.0514 \pm 0.0232$ | $0.254 \pm 0.028$ | $0.0486 \pm 0.0265$ |
| 0.4 | $0.0667 \pm 0.0279$ | $0.218 \pm 0.063$ | $0.0367 \pm 0.0267$ | $0.0459 \pm 0.0251$ | $0.273 \pm 0.026$ | $0.0541 \pm 0.0308$ |
| 0.45 | $0.0333 \pm 0.0279$ | $0.212 \pm 0.068$ | $0.03 \pm 0.0245$ | $0.0486 \pm 0.0369$ | $0.276 \pm 0.025$ | $0.0541 \pm 0.032$ |

| Cost | haberman TSB | haberman DHL | haberman BA | pima TSB | pima DHL | pima BA |
|---|---|---|---|---|---|---|
| 0.05 | $0.0 \pm 0.0$ | $0.000 \pm 0.000$ | $0.0 \pm 0.0$ | $0.0013 \pm 0.0026$ | $0.023 \pm 0.052$ | $0.0 \pm 0.0$ |
| 0.1 | $0.00323 \pm 0.00645$ | $0.069 \pm 0.042$ | $0.0 \pm 0.0$ | $0.013 \pm 0.0109$ | $0.145 \pm 0.016$ | $0.0 \pm 0.0$ |
| 0.15 | $0.0 \pm 0.0$ | $0.161 \pm 0.054$ | $0.0452 \pm 0.0903$ | $0.00779 \pm 0.00486$ | $0.196 \pm 0.026$ | $0.109 \pm 0.0267$ |
| 0.2 | $0.0161 \pm 0.0144$ | $0.203 \pm 0.041$ | $0.171 \pm 0.0332$ | $0.0143 \pm 0.00954$ | $0.222 \pm 0.023$ | $0.106 \pm 0.0106$ |
| 0.25 | $0.0 \pm 0.0$ | $0.246 \pm 0.026$ | $0.187 \pm 0.0347$ | $0.0117 \pm 0.00757$ | $0.227 \pm 0.025$ | $0.127 \pm 0.0286$ |
| 0.3 | $0.0226 \pm 0.0194$ | $0.259 \pm 0.021$ | $0.168 \pm 0.0416$ | $0.00779 \pm 0.00636$ | $0.235 \pm 0.023$ | $0.122 \pm 0.0161$ |
| 0.35 | $0.00968 \pm 0.0129$ | $0.259 \pm 0.027$ | $0.245 \pm 0.0524$ | $0.0013 \pm 0.0026$ | $0.236 \pm 0.024$ | $0.136 \pm 0.025$ |
| 0.4 | $0.0226 \pm 0.0219$ | $0.259 \pm 0.027$ | $0.252 \pm 0.0573$ | $0.00649 \pm 0.0101$ | $0.242 \pm 0.029$ | $0.142 \pm 0.0199$ |
| 0.45 | $0.0258 \pm 0.0079$ | $0.252 \pm 0.025$ | $0.252 \pm 0.0573$ | $0.00779 \pm 0.00636$ | $0.248 \pm 0.036$ | $0.151 \pm 0.0226$ |

| Cost | australian TSB | australian DHL | australian BA | banknote TSB | banknote DHL | banknote BA |
|---|---|---|---|---|---|---|
| 0.05 | $0.0 \pm 0.0$ | $0.104 \pm 0.033$ | $0.0391 \pm 0.00983$ | $0.000727 \pm 0.00145$ | $0.051 \pm 0.061$ | $0.00218 \pm 0.00178$ |
| 0.1 | $0.0058 \pm 0.00542$ | $0.113 \pm 0.023$ | $0.0449 \pm 0.0141$ | $0.00218 \pm 0.00291$ | $0.074 \pm 0.070$ | $0.00655 \pm 0.00626$ |
| 0.15 | $0.0145 \pm 0.00648$ | $0.120 \pm 0.023$ | $0.0536 \pm 0.0126$ | $0.000727 \pm 0.00145$ | $0.072 \pm 0.076$ | $0.00364 \pm 0.00398$ |
| 0.2 | $0.0145 \pm 0.0102$ | $0.123 \pm 0.037$ | $0.0754 \pm 0.0208$ | $0.000727 \pm 0.00145$ | $0.034 \pm 0.017$ | $0.00509 \pm 0.00493$ |
| 0.25 | $0.0101 \pm 0.0058$ | $0.126 \pm 0.037$ | $0.0841 \pm 0.0087$ | $0.0 \pm 0.0$ | $0.045 \pm 0.018$ | $0.00509 \pm 0.00371$ |
| 0.3 | $0.0058 \pm 0.0029$ | $0.128 \pm 0.036$ | $0.0942 \pm 0.02$ | $0.00218 \pm 0.00291$ | $0.066 \pm 0.021$ | $0.00582 \pm 0.00371$ |
| 0.35 | $0.0087 \pm 0.0029$ | $0.132 \pm 0.036$ | $0.104 \pm 0.0187$ | $0.00145 \pm 0.00178$ | $0.072 \pm 0.044$ | $0.00655 \pm 0.00424$ |
| 0.4 | $0.0145 \pm 0.0112$ | $0.143 \pm 0.041$ | $0.145 \pm 0.0271$ | $0.00218 \pm 0.00178$ | $0.100 \pm 0.028$ | $0.00582 \pm 0.00436$ |
| 0.45 | $0.00435 \pm 0.0058$ | $0.148 \pm 0.044$ | $0.109 \pm 0.0314$ | $0.00218 \pm 0.00291$ | $0.112 \pm 0.024$ | $0.00509 \pm 0.00371$ |