[Reviews · NeurIPS 2016]

Reviewer 1

Summary

The paper studies in depth the problem of boosting when the classifier is permitted to abstain, but at a cost. The paper develops theory, an algorithm, and runs experiments.

Qualitative Assessment

This paper carries through a thorough examination of this problem. The problem is interesting and valuable. The theoretical results are decent. The experiments seem well done. The algorithm looks reasonable. Perhaps there are no surprises either in the results or methods used, but the work is nevertheless worthwhile and nicely done. The presentation is mostly excellent, clear and polished. It seems that the loss function defined in (1) transforms the problem into a kind of cost-senstive, multiclass learning problem. To be more concrete, we can view the problem as one in which the data has labels -1 or +1, but the classifier can predict with three values: -1, +1 or abstain. The loss is then c (or c(x)) if the classifier abstains; 0 if the classifiers label matches the true label; and 1 otherwise. In other words, we get a matrix of losses for each possible true/predicted label combination. At this point, it seems to me that a general-purpose algorithm for cost-sensitive, multiclass learning or boosting could be used. For instance, a natural candidate would be one of the algorithms in Mukherjee et al's "Theory of Multiclass Boosting" paper, which I believe can handle cost-sensitive problems. So the question is: How would such an algorithm compare to the one in the paper? For instance, can the one in the paper be viewed as a special case of one of these algorithms? What are the advantages of each compared to the other? It would definitely be helpful to explain the meaning and significance of Theorem 2 and Corollary 3. Regarding the algorithm in Section 4, I wonder if there is also a natural algorithm that does not explicitly regularize, and that allows the alpha_t's to be unbounded, similar to AdaBoost. Detailed comments: line 60: must c(x) be known to the learner? line 102: why must H and R be distinct? line 108: The notation H_k and R_k have not been defined. And is k_t an integer? The notation makes it look like a real number, which doesn't make sense. line 109: What are "base functions" in this context? line 114: Saying that F(-u) is non-increasing is kind of confusing because of the reversal of sign. Why not just say that F is non-decreasing? line 131: Clearer to say "strictly decreasing" rather than "strictly non-increasing". line 140: Need to say what R_m is.

Confidence in this Review

3-Expert (read the paper in detail, know the area, quite certain of my opinion)


Reviewer 2

Summary

The papers considers a new boosting algorithm for classification with abstention. In each round, the algorithm selects a a base learner which is a pair of functions (h(x),r(x)), where h concerns classification into positive/negative labels, while r indicates the abstention region. The paper contains extensive theoretical analysis of the proposed approach, including: convex surrogates for the proposed loss function, calibration of exponential-loss convex surrogate, generalization bounds by means of Rademacher complexity, derivation of the boosting algorithm and its convergence guarantees. A short experimental study concludes the paper.

Qualitative Assessment

The paper presents an interesting, and (to the best of my knowledge) novel approach to binary classification with abstention. The standard approach to this problem is via confidence-based abstention: "estimate" the class conditional probability function \eta(x) (or some monotonic transformation thereof) and reject (abstain from classification) when \eta(x) is close to 1/2. The authors points out that while this is all that is needed to find the optimal solution in the class of all classification functions (because the Bayes solution is of that form), in practical cases, when the class of functions is constrained, it can be beneficial to use a more general approach, where there are two functions, h and r, h being responsible for classification, and r -- for deciding upon rejection. The paper contains various theoretical results concerning the proposed approach: - Convex surrogates for the original (task) loss function, with the proof of calibration of exponential convex surrogate, - Guarantees on the generalization error of the ensemble by means of the convex surrogate and the Rademacher complexity of the function class which depends on the Lipschitz constant of the convex surrogate (truncated at 1), - Derivation of new boosting algorithms based on the proposed convex surrogates using the method of projected coordinate descent, - Proof of the convergence of the projected coordinate descent, - A new type of based learner suitable for this problem -- abstention stumps, together with a method for fast selection of the optimal stump at each iteration. - A short experimental study, which is actually the weakest part of the paper due to small size of the data sets used for the comparison. The paper is generally clearly presented, but due to numerous theoretical results, it is quite dense. I have some minor suggestions on the presentation (below) that may help the final version. Overall, I think the contribution of the paper is strong. I am in favor of accepting it. Minor suggestions: l. 68: "part of our analysis is applicable to the general case" -- please explain which part. l. 82-84 and Fig. 1: the Bayes solution is not unique here. l. 98-99: repeated sentence ("ham based on the presence of some other word") l. 60, 69, 105: unify notation for open and closed intervals eq. between lines 113-114: h(x)) -> h(x) l. 121: I think it better to write \Phi_1(x) = \Phi_2(x) = \exp(x) Theorem 1: it is not clear what are h^* and r^* -- is (h^*,r^*) a Bayes classifier for the task loss L? Also the proof of Theorem 1 is unclear. In l. 467 (Appendix): in fact when \eta(x) \in {0,1}, there is no minimizer (for example, when \eta(x)=1, the infimum is attained by r(x) -> infinity and h(x) - r(x) -> infinity). Furthermore, in l. 476-477 (Appendix) you write that the Bayes classifier (of the task loss L?) satisfies h^*(x) = \eta(x) - 1/2 and r^*(x) = |h^*|-1/2+c, but this is not true. In fact there exists Bayes classifier which satisfies that, but this is not the only one: any (h^*, r^*) is a Bayes classifier as long as sgn(h^*) = sgn(eta(x)) and sgn(r^*) = sgn(|\eta(x)-1/2|-1/2-c). This is satisfied for the solution of the surrogate if and only if b/a = 2sqrt((1-c)/c), so Theorem 1 is correct, but I think the proof should be improved. Theorem 2: explain the notation -- R is the Rademacher complexity (define it in the Appendix to make it more self-contained). Theorem 2: Perhaps you could add a comment below Thm 2 that it also applies to L_SB since it is an upper bound on L_MB (similar to the comment below Corollary 3) Figure 3: (algorithm) It is unclear, what are the parameters of the algorithm. From what I understand, they are c and \beta? I would suggest to say anything about the role of \beta in the algorithm (from what I understand, it acts as a regularization strength). Experiment: it is not explicitly stated what kind of convex surrogate is used in the BA algorithm. I guess this is exactly the algorithm presented on Figure 3 (with exponential loss, L_SB-type). ========== I have read the rebuttal which clarified most of the issues raised in my review.

Confidence in this Review

2-Confident (read it all; understood it all reasonably well)


Reviewer 3

Summary

The paper proposes a novel boosting algorithm which aims to minimise the abstention loss, which for each instance charges either the fixed cost c if the model abstains, or the cost of 1 if the model misclassifies. The majority of the paper works on deriving suitable convex surrogate losses for the abstention loss and on deriving the boosting algorithm for these losses. Towards the end it specialises the method to use abstention stumps, and proposes a method for learning one such stump faster than the naive method. Finally, the method is compared experimentally against 2 existing methods on 6 datasets, showing favourable performance of the proposed BA algorithm.

Qualitative Assessment

This is an interesting work on a very relevant topic. The authors should be praised for the effort of taking on this challenge of deriving suitable convex surrogate losses and deriving the boosting algorithm, as this required quite a lot of technical theoretical work. In particular, I like the guarantee from Theorem 1 about when the surrogate loss is a suitable one, and the fact that the BA algorithm gives L1 regularised AdaBoost if the rejection part is eliminated. I have three main concerns with this paper. The first is the low number of datasets (6) in the experiments, and perhaps also that the datasets are so small (biggest has 1370 instances), and perhaps that there are only 2 algorithms compared against. The experiments show that the method does work reasonably well on these datasets, but do not convince the reader that it is with high probability significantly better on a new dataset. Perhaps this is ok, but still it should not have been hard to double the number of datasets. My second concern is the very limited discussion of experimental results. The supplementary material has interesting tables of average fraction of points rejected and average classification error on non-rejected points, but these have not been discussed. My third, smaller concern is that quite some important material has been pushed to the supplementary, such as the extended related work, descriptions of 2 competing algorithms, and the descriptions of datasets. Together with the previous concern I think this highlights the main problem of the paper: too much material for the NIPS format. This would have made a much better journal paper. Related to the shortage of space, the paper simply states in line 196 that if the abstention part is eliminated from the BA algorithm then what remains is L1-regularised AdaBoost. I don't find this to be obvious at all, looking at the complicated algorithm in Figure 3. This surely requires more explanation. As a minor issue, in line 104 the introduction of gamma should be complemented with an explanation that gamma is introduced to allow tuning of the rejection threshold. A minor suggestion, I see the order r,h much more natural than h,r because the model h only becomes relevant if r does not reject (e.g. on line 74). Also, the term 'abstention loss' is perhaps a bit misleading, suggesting that it only accounts for the cost of abstention, rather than combining the losses of abstention and misclassification.

Confidence in this Review

1-Less confident (might not have understood significant parts)


Reviewer 4

Summary

The paper proposed new boosting algorithm for binary classification with abstention. The algorithms was compared with confidence-based algorithms DHL and TSB.

Qualitative Assessment

The experiments in the paper are constraints to a small set of datasets from UCI, sample size and number of features are too small. In order to make the results more convincing, the authors should consider some larger datasets.

Confidence in this Review

2-Confident (read it all; understood it all reasonably well)


Reviewer 5

Summary

The authors study a boosting classifier with abstention. They find simultaneously the classifier and the rejection function which is of a general form (and not necessarily a threshold type as studied in the past). A boosting algorithm is presented that concurrently finds both functions. The experiments show the superiority of the joint approach.

Qualitative Assessment

I believe this is an important and non-trivial generalization of boosting with abstention. I like the main idea and then everything leading to the algorithm. An unclear point: the linear combination of classifiers and rejection function is the same, i.e, alphas are used for both. In a more general setting there would be alphas for classification functions and betas for rejection functions. Having alpha=beta might with WLOG but if this is the case, it should be clarified.

Confidence in this Review

2-Confident (read it all; understood it all reasonably well)


Reviewer 6

Summary

This paper present a new boosting algorithm for binary classification with abstention. At each round, simultaneously selects a pair of functions, a base predictor and a base abstention function. This paper also report the results of several experiments comparing BA to two different confidence-based algorithms (TSB, DHL), which suggest that BA provides a significant improvement in practice.

Qualitative Assessment

This work has rigorous proofs and very good results include on CIFAR10 dataset (boat and horse images). It is a very outstanding work.

Confidence in this Review

1-Less confident (might not have understood significant parts)